# PaLD: Detection of Text Partially Written by Large Language Models

**Eric Lei**[12][*] **Hsiang Hsu**[2]**, Chun-Fu (Richard) Chen**[2]
[1]University of Pennsylvania, [2]JPMorganChase Global Technology Applied Research
elei@seas.upenn.edu, {hsiang.hsu, richard.cf.chen}@jpmchase.com

## Abstract

Advances in large language models (LLM) have produced text that appears increasingly human-like and difficult to detect with the human eye. In order to mitigate the impact of misusing LLM-generated texts, e.g., copyright infringement, fair student assessment, fraud, and other societally harmful LLM usage, a line of work on detecting human and LLM-written text has been explored. While recent work has focused on classifying entire text samples (e.g., paragraphs) as human or LLM-written, this paper investigates a more realistic setting of *mixed-text*, where the text's individual segments (e.g., sentences) could each be written by either a human or an LLM. A text encountered in practical usage cannot generally be assumed to be fully human or fully LLM-written; simply predicting whether it is human or LLM-written is insufficient as it does not provide the user with full context on its origins, such as the amount of LLM-written text, or locating the LLM-written parts. Therefore, we study two relevant problems in the mixed-text setting: (i) estimating the percentage of a text that was LLM-written, and (ii) determining which segments were LLM-written. To this end, we propose *Partial-LLM Detector* (PaLD), a black-box method that leverages the scores of text classifiers. Experimentally, we demonstrate the effectiveness of PaLD compared to baseline methods that build on existing LLM text detectors.

## 1 Introduction

Large language models (LLMs) have demonstrated capabilities to generate text that convincingly impersonates humans[1] (Achiam et al., 2023; Floridi & Chiriatti, 2020; Chowdhery et al., 2023). In conjunction with their wide deployment and easy accessibility, these LLMs have posed potential risks across industries and society. Namely, LLM-generated text may contaminate the development of next-generation foundation models (Shumailov et al., 2024; 2023), facilitate the spread of fake or biased content (Bender et al., 2021; Farina & Lavazza, 2023; Li et al., 2023), unintentionally infringe on copyrights (Mitchell & Krakauer, 2023), and impair education by depriving students of the effort needed to compose their own articles (Cotton et al., 2024). The potential downsides of LLMs, particularly in scenarios where humans can be easily deceived by text generated by these models, underscore the need for reliable methods to audit and detect LLM-generated content.

Given an article, such as a paragraph, existing methods for detecting LLM-generated text are often cast as a binary classification problem, i.e., assigning a binary label to the article to indicate whether it was written by a human or generated by an LLM (Gehrmann et al., 2019; Ippolito et al., 2020; Mitchell et al., 2023; Mao et al., 2024; Verma et al., 2024). However, the binary classification approach may fail to provide finer-grained information about the extent of LLM involvement in composing the article, as it may not be entirely written by LLMs. In fact, recent studies have shown that LLMs are frequently used to edit, refine, or rephrase *only parts* of an article (Črček & Patekar, 2023; Levine et al., 2024)—see Fig. 1 for a concrete example. Moreover, when using existing methods to detect text that is partially written by LLMs, they may either exhibit excessive confidence in identifying the whole article as LLM-generated content, or be overly conservative

---

[*]Work done while an intern at JPMorganChase.
[1]Recent works reported that LLM-generated text is difficult to detect with the human eye (Mei et al., 2024; Gehrmann et al., 2019; Guo et al., 2023), and could conditionally pass the Turing test (Jones & Bergen, 2023).

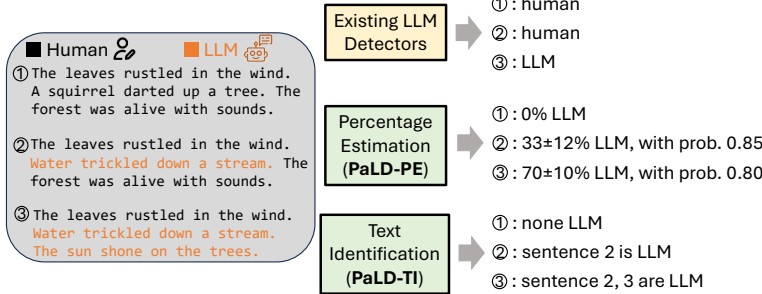

Figure 1: In practice, text encountered may be partially-LLM written. Existing LLM detectors can only predict whether the text is human- or LLM-written. In contrast, our method (PaLD) enables LLM percentage estimation and LLM text identification. PaLD-PE provides confidence intervals on the amount of LLM text, and PaLD-TI provides a likelihood of each sentence being LLM.

towards classifying it as human writing, as empirically demonstrated by text 2 and text 1 in Fig. 2, respectively. Incorrectly assessing the level of LLM intervention can lead to unjust penalties for light LLM usage, or encourage unauthorized usage by those who exploit these inaccuracies. These practical concerns, therefore, motivate the need to refine the detection techniques in existing methods.

In this paper, we address a more realistic setting of *mixed-text*, where a piece of text to be audited consists of both LLM-generated and human-written content. This mixed-text setting leads to two primary research goals: (i) *percentage estimation*, which aims to estimate the proportion of text in an article that was generated by LLMs, and (ii) *LLM text identification*, which seeks to identify the specific text segments in the article that are more likely to have been generated by LLMs. Existing LLM-text detectors typically design a statistic of the LLMs' outputs given the article, which we refer to as the $T$-score, to detect distribution shifts between texts that are either fully written by LLMs or by humans, and then threshold the $T$-score for a binary decision. However, the mixed-text setting requires a more sensitive statistic to detect subtle distribution shifts, such as when the text is composed of 20% human text and 80% LLM text.

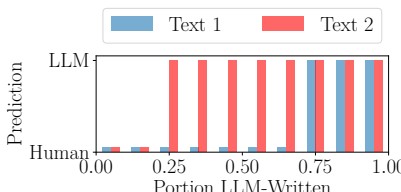

Figure 2: Binary classification of two human texts progressively overwritten by LLM. RoBERTa (Verma et al., 2024) predicts the entire text 1 as human-written when up to 75% of it is LLM-written; only 25% of text 2 is LLM-written before RoBERTa classifies the entire text as LLM.

To address this technical challenge, we develop a Bayesian framework based on the $T$-score, termed the Partial-LLM Detector (PaLD), which provides maximum a posteriori (MAP) estimates of the percentage of LLM-generated text in an article (Cohen, 2019). Using mixture Gaussian kernel density estimation (Sharif-Razavian & Zollmann, 2008; Gelman et al., 2004), we further derive credible intervals that reliably cover the ground-truth percentage. For LLM text identification, the PaLD framework is statistically more robust, especially when each text segment in an article is relatively short, where existing LLM detectors lack sensitivity in detecting the distributional shift due to the short text length. We formulate a set optimization problem that searches for the set of segments that maximize the discrepancy of the $T$-scores between the set and its complement. This optimization problem can either be exactly solved or effectively approximated by a greedy policy, flagging the set of text segments that are more likely to be LLM-generated. We term the PaLD framework for solving the two research goals as PaLD-PE (for percentage estimation) and PaLD-TI (for LLM text identification), respectively. Fig. 1 summarizes the PaLD framework, along with its use cases.

The rest of this paper is organized as follows. In Section 2, we define notations, provide the mathematical background, and survey related work. Section 3 formally introduces the PaLD framework and demonstrates how to perform PaLD for percentage estimation (PaLD-PE) and LLM text identification (PaLD-TI) with statistical guarantees. In Section 4, we empirically illustrate that PaLD-PE and PaLD-TI outperform existing detection methods on two language datasets: WritingPrompts (Fan et al., 2018) and Yelp Reviews (Yelp, 2014). Finally, we present our concluding remarks, including limitations and future directions, in Section 5. Code to reproduce our experiments can be accessed at https://github.com/jpmorganchase/pald.

## 2    BACKGROUND AND RELATED WORK

Let $X$ a random variable of text that follows a text-generation distribution $P$, which can either be human writing ($P_{\text{human}}$) or LLMs ($P_{\text{LLM}}$), and let $x \in \mathcal{X}$ be a realization from $P$. Here, $\mathcal{X}$ is the sample space of texts; for example, $\mathcal{X} = \mathcal{D}^*$ would contain all finite-length strings drawn from a dictionary $\mathcal{D}$. The goal of LLM text detection is to design a statistic, referred to as the $T$-score (short for text score), that maps a piece of text to a scalar value, i.e., $T : \mathcal{X} \to \mathbb{R}$. The desired property of the $T$-score is that the distributions of $T(X)$ when evaluated with $X \sim P_{\text{human}}$ and $X \sim P_{\text{LLM}}$ result in two statistically separated modes. A text sample $x$ can then be classified as LLM if $T(x) > \gamma$, where $\gamma$ is a preset threshold.

$T$**-scores in LLM text detection.** Existing LLM detectors design various $T$-scores to improve classification performance. Typically, $T$-scores can either be inferred from pre-trained models, as demonstrated by Mitchell et al. (2023) with DetectGPT, explicitly learned for binary classification tasks in models like RoBERTa (Guo et al., 2023) or Ghostbuster (Verma et al., 2024), or through a combination of both approaches, such as in RAIDAR (Mao et al., 2024). For example, Solaiman et al. (2019) and Ippolito et al. (2020) utilize the average log-probability of a text as the $T$-score. DetectGPT, on the other hand, calculates log-probability curvature, defined as $T(x) = \log P_{\text{LLM}}(x) - \mathbb{E}_{\tilde{x}}[\log P_{\text{LLM}}(\tilde{x})]$, where $\tilde{x}$ represents a perturbed version of $x$ generated by Google's T5 (Raffel et al., 2020). This method was later extended to conditional log-probability curvature by Bao et al. (2024) with Fast-DetectGPT, and further explored by Mireshghallah et al. (2024).

$T$-scores can also be defined using the logits from LLMs, which are typically thresholded at $\gamma = 0$ for classification tasks, and then used to train a binary classifier (Sadasivan et al., 2023). For instance, Guo et al. (2023) and Chen et al. (2023) fine-tune RoBERTa models (Liu et al., 2019) to differentiate between LLM-generated and human-written texts. Verma et al. (2024) enhances the generalization performance of their models by employing logistic regression classifiers on selected features from LLM token probabilities. Conversely, Mao et al. (2024) introduce a new $T$-score based on the rewriting Levenshtein score, opting for this over logits in their tree-based classification method. Finally, Gehrmann et al. (2019) use the ranking of the top-$k$ log probabilities as the $T$-score, noting that human-written texts tend to be sampled more frequently from the tail of a LLM's probability distribution. Although $T$-scores are designed for binary classification, our work shows how they can be leveraged for the percentage estimation and text identification tasks that we propose.

**LLM boundary detection.** A few recent works have investigated detecting the boundary when text goes from human-written to LLM-written. The RoFT dataset (Dugan et al., 2020), containing human-written sentences completed by GPT2, was created to evaluate how humans detect the boundary. Cutler et al. (2021); Clark et al. (2021); Zeng et al. (2023); Wang et al. (2023) provide RoBERTa or Transformer-based models for solving the task in an automated fashion, with Wang et al. (2023) requiring white-box access to the ground-truth LLM. Kushnareva et al. (2024) evaluates several approaches and finds that perplexity-based approaches are more robust for boundary detection. Our proposed framework, PaLD, is more general, as it encompasses settings where any segment of a text could be human or LLM-written, does not require white-box access to a LLM, and can be used with both supervised $T$-scores as well zero-shot $T$-scores.

**LLM watermarking and hallucination.** Techniques used to detect distribution shifts in language generation are also applicable in related fields. For instance, LLM watermarking embeds a unique pattern, or signature, within the output distributions of an LLM to safeguard its authorship (Kirchenbauer et al., 2023; Kamaruddin et al., 2018; Zhao et al., 2024). This signature induces a distinct distribution shift in the text, making it uniquely identifiable and easily distinguishable from outputs generated by other LLMs. Despite sharing similar techniques, LLM watermarking serves a fundamentally different purpose compared to LLM text detection, which is the setting of this paper.

## 3    PALD: PARTIAL-LLM DETECTOR

We start with the formulation of the mixed-text setting by splitting a piece of text $x$ into a concatenation of $n$ segments, i.e., $x = x_1 \ldots x_n$. We assume that each segment $x_i$ has either been generated by LLMs ($x_i \sim P_{\text{LLM}}$), or written by a human ($x_i \sim P_{\text{human}}$). For example, consider the following paragraph $x = x_1 x_2 x_3 x_4$, decomposed into $n = 4$ segments:

| $T$-score | $\sigma_T^*$ |
|---|---|
| RoBERTa-LN | 2.599 |
| RoBERTa | 2.160 |
| Ghostbuster (Verma et al., 2024) | 1.889 |
| FastDetectGPT (Bao et al., 2024) | 1.784 |
| DetectGPT (Mitchell et al., 2023) | 1.496 |

Table 1: Normalized quantile slope in (1) for different $T$-scores. Higher is better.

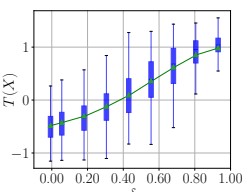

Figure 3: Distribution shift of RoBERTa-LN with varying $\delta$.

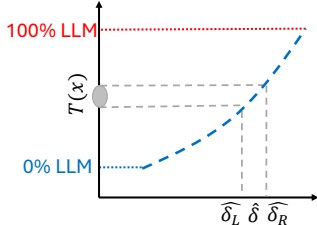

Figure 4: PaLD-PE for percentage estimation returns point $\hat{\delta}$ and interval $(\widehat{\delta_L}, \widehat{\delta_R})$ estimates.

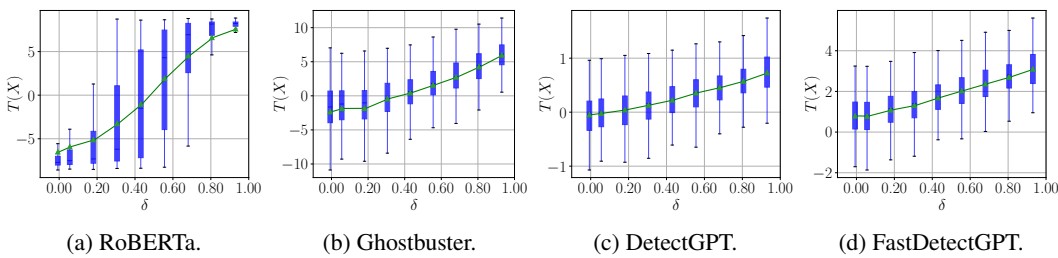

(a) RoBERTa.    (b) Ghostbuster.    (c) DetectGPT.    (d) FastDetectGPT.

Figure 5: Distribution shift of various $T$-scores with mixed-text fraction $\delta$.

> It was an excellent decision. The pancakes were fluffy and bursting with blueberries. They layered the cheese on the sandwich, which to me is a must for a true egg sandwich. The meal also came with a side of breakfast potatoes.

Black segments (i.e., sentences) $x_1$ and $x_3$ are human-written, while red segments $x_2$ and $x_4$ are LLM-generated. It is noteworthy that existing LLM text detectors typically operate in the setting where that all segments $\{x_i\}_{i=1}^n$ are either human-written or LLM-generated.

### 3.1 ADOPTING $T$-SCORES FOR MIXED TEXTS

In the binary classification setting, the $T$-score is used to discriminate fully-human texts $X_{\text{human}} \sim P_{\text{human}}$ and fully-LLM texts $X_{\text{LLM}} \sim P_{\text{LLM}}$. Here, there are only two distributions of interest, and the $T$-score merely has to discriminate their samples. Recent work such as Sadasivan et al. (2023) shows that the best $T$-score (in terms of simultaneously reducing the true and false positive rates) is one that separates $T(X_{\text{human}})$ and $T(X_{\text{LLM}})$ as much as possible. In the mixed-text setting, the $i^{\text{th}}$ segment $X_i$ in a random mixed text $X$ follows the mixed distribution, i.e., $X_i \sim P_{\text{mixed}} \triangleq \eta P_{\text{LLM}} + (1-\eta)P_{\text{human}}$, where $0 \le \eta \le 1$ represents the segment-level LLM fraction. As each segment may not have the same number of characters, we further define $\delta \in [0,1]$ as the character-level[2] LLM fraction of $X$. The mixed distribution reverts to $P_{\text{human}}$ when $\eta = 0$ (thereby making $\delta = 0$), and recovers $P_{\text{LLM}}$ when $\delta = 1$. As $\delta$ increases, the mixed distribution of $X$ shifts from $P_{\text{human}}$ to $P_{\text{LLM}}$.

Assessing the amount of LLM-generated text within $X$ equates to estimating $\delta$. Ideally, as $\delta$ varies, the distribution shift in $T(X)$ should smoothly transition between $T(X_{\text{human}})$ and $T(X_{\text{LLM}})$. However, accurate estimation of $\delta$ using the $T$-score becomes challenging if there is significant overlap among the $T$-score distributions for different $\delta$ values, as multiple $\delta$ values could plausibly explain the same $T$-score. The degree of overlap can be quantified by the normalized $T$-score quantile slope, $\sigma_T(p)$, defined as follows:

$$\sigma_T(p) \triangleq \int_{[0,1]} \frac{\frac{d}{d\delta}Q(\delta, p)}{Q(\delta, 1 - \frac{\epsilon}{2}) - Q(\delta, \frac{\epsilon}{2})} d\delta, \tag{1}$$

where $Q(\delta, p) \triangleq \arg\min_t \Pr(T(X) \le t) = p$ is the quantile function for the $T$-score distribution at $\delta$. The target is to get $\sigma_T^* \triangleq \min_{\frac{\epsilon}{2} \le p \le 1 - \frac{\epsilon}{2}} \sigma_T(p)$ as large as possible. Here, $\epsilon$ adjusts the considered

---

[2]Using the previous example, $\eta = 0.5$ since LLM generates two sentences. However, there are 109 out of 197 LLM-generated characters, and therefore $\delta = 109/197 \approx 0.553$.

quantile range. A high value of $\sigma_T^*$ ensures that $\delta$ can be uniquely identified across its entire range. Conversely, a low $\sigma_T^*$ indicates substantial overlap in the $T$-score distributions for different $\delta$ values, complicating the unique identification of $\delta$.

We conclude this subsection with an empirical example. We first prepare the mixed text by a mask-and-fill approach. Consider a case where $n = 5$ using human-written text $x_{\text{human}} = x_1 x_2 x_3 x_4 x_5$. Setting $\eta = 0.4$, we randomly mask two sentences with the special token [MASK]. This modifies the text to, for example, $x_{\text{LLM}} = x_1$[MASK]$x_3 x_4$[MASK]. We then prompt GPT-4o to fill in the [MASK] tokens in $x_{\text{LLM}}$, resulting in a mixed text sample with $\eta = 0.4$. Further details on this approach are provided in Section A.1. With the method to generate the mixed text by $\eta$ (equivalently $\delta$), in Tab. 1, we compare the $T$-scores of several methods using $\sigma_T^*$. We find that the logits with RoBERTa classifier trained with LogitNorm loss (Wei et al., 2022) to classify fully-human/LLM text yield the best $\sigma_T^*$; we refer to this method as RoBERTa-LN. The $T$-score distribution shift with varying $\delta$ is reported in Fig. 3 for RoBERTa-LN as well as the other $T$-scores listed in Fig. 5. It can be seen that all methods achieve good statistical separation between $T(X_{\text{human}})$, i.e. $\delta = 0$, and $T(X_{\text{LLM}})$, i.e., $\delta = 1$. As will be shown in an ablation study in Sec. 4, the performance of PaLD for percentage estimation and text identification highly correlate with the $\sigma_T^*$ values of the $T$-scores used. Next, we demonstrate how the $T$-score in the mixed-text setting can be used to for PaLD-PE and PaLD-TI.

## 3.2 Percentage Estimation

Given a mixed-text realization $x$ with ground-truth LLM fraction $\delta$, we would like to produce either a point estimate $\widehat{\delta}$ of $\delta$, or a predictive interval $(\widehat{\delta_L}, \widehat{\delta_R})$ that contains $\delta$ with high probability. A predictive interval provides the user with a measure of confidence on the estimated percentage value of the mixed text $x$. PaLD-PE uses a Bayesian approach to estimate $\delta$, and thus assumes it is random; we denote the random LLM fraction as $\Delta$. At a high-level, PaLD-PE first estimates the joint statistics between the LLM text percentage and the $T$-scores, then uses this model to return point estimates and/or predictive intervals of $\delta$; see Fig. 4. In the first step, mixed texts are generated from a fully-human dataset to measure the shift in distribution of the text score $T$ as the LLM percentage ranges from 0 to 1; we then fit a mixture kernel density estimate (KDE) to estimate the likelihood $P(T(X)|\Delta)$. In the second step, when we estimate the LLM percentage of an unseen text sample, we use the posterior $P(\Delta|T(X))$ to return maximum a posteriori (MAP) estimates for $\widehat{\delta}$ and highest density intervals (HDI) for $(\widehat{\delta_L}, \widehat{\delta_R})$.

**Measuring the $P(T(X)|\Delta)$ likelihood.** To model $P(T(X)|\Delta)$, we need to gather pairs of samples representing $\Delta$ and $T(X)$. For $X$, we first generate synthetic mixed texts by using an LLM to fill randomly masked out sentences of fully-human texts (Sec. A.1). These synthetic mixed texts are generated at $K$ target fraction levels $0 \leq \delta_1 < \cdots < \delta_K \leq 1$. Let $\{x_i^{(k)}\}_{i=1}^{n_k}$ denote the mixed texts generated at target fraction $\delta_k$. We compute the $T$-score of each such text, yielding $\{t_i^{(k)}\}_{i=1}^{n_k}$, where $t_i^{(k)} = T(x_i^{(k)})$. Thus, each $t_i^{(k)}$ is assumed drawn from the $P(T(X)|\Delta = \delta_k)$ distribution.

Now, we parameterize a model for the likelihood $P(T(X)|\Delta)$, using a mixture of KDEs. An individual KDE is fit for each conditional $P(T(X)|\Delta = \delta_k)$. For the full conditional $P(T(X)|\Delta)$, we take convex combinations of the nearest two KDEs to $\Delta$. Specifically, let $\phi_k(t) = \frac{1}{n_k} \sum_{i=1}^{n_k} \frac{1}{h} K\left(t - t_i^{(k)}/h\right)$ be a Gaussian KDE fit to the samples $\{t_i^{(k)}\}_{i=1}^{n_k}$. Here, $K(z) = \frac{1}{\sqrt{2\pi}} e^{-\frac{1}{2} z^2}$, $n_k$ is the number of $T$-score samples collected at $\delta_k$, and $h$ is a bandwidth parameter. Then, our mixture KDE computes the likelihood as

$$P(T(X) = t | \Delta = \delta') = \theta \phi_{k^*}(t) + (1 - \theta) \phi_{k^*+1}(t), \tag{2}$$

where $k^*$ is the index such that $\delta_{k^*} \leq \delta' < \delta_{k^*+1}$, and $\theta = \frac{\delta_{k^*+1} - \delta'}{\delta_{k^*+1} - \delta_{k^*}}$.

**Percentage prediction.** To predict the LLM fraction, we use the posterior density $P(\Delta|T(X)) \propto P(T(X)|\Delta)P(\Delta)$, where we assume a prior distribution $P(\Delta)$ supported on $[0, 1]$. Let $x$ be the text sample we would like to estimate the LLM percentage. For the point estimate, we return the MAP estimate $\hat{\delta} = \arg\max_{\delta'} P(\Delta = \delta' | T(X) = T(x))$. For the predictive interval, we return the

Figure 6: PaLD-TI for LLM text identification. Left: stitched texts $x[S]$ are enumerated, and $T$-scores computed. Middle: $f_x(S)$, the $T$-score difference, is computed for all $S$. We illustrate this for $S = \{1\}$. Right: the maximum $f_x(S)$ is computed, and its maximizing set $\hat{S}$ is returned as the segment indices predicted as LLM.

$(1 - \alpha)$-HDI (Chen & Shao, 1999),

$$
\begin{aligned}
\delta_L, \delta_R = \underset{\delta_L, \delta_R}{\arg\max} \quad & \pi_\alpha \\
\text{s.t.} \quad & P(\delta_L \leq \Delta < \delta_R | T(X) = T(x)) \geq \pi_\alpha \\
& P(\delta_L \leq \Delta < \delta_R | T(X) = T(x)) \geq 1 - \alpha,
\end{aligned}
\tag{3}
$$

where $\alpha$ is a parameter we can set to control the posterior probability that $\delta$ is contained in the interval. In practice, since we do not have the exact posterior density, we use a Markov Chain Monte Carlo approach (Chen & Shao, 1999) by sampling $\delta'_1, \ldots, \delta'_M$ from the posterior $P(\Delta | T(X) = T(x))$ via Metropolis-Hastings (Gelman et al., 2004). For the MAP estimate, we return the sample mode $\hat{\delta} = \arg\max_{1 \leq i \leq M} P(\Delta = \delta'_i | T(X) = T(x))$. For the $(1 - \alpha)$-HDI estimate, we return $(\widehat{\delta_L}, \widehat{\delta_R}) = (\delta'_{(i^*)}, \delta'_{(i^* + [(1-\alpha)M])})$, where $\delta'_{(i)}$ is the $i$-th smallest sample, and $i^* = \arg\min_{1 \leq i \leq M} \delta'_{(i^* + [(1-\alpha)M])} - \delta'_{(i^*)}$.

## 3.3 LLM TEXT IDENTIFICATION

The goal is to return an index set $\hat{S} \subseteq \{1, \ldots, n\}$ corresponding to a segmentation $x = x_1 \ldots x_n$ such that $\{x_i : i \in \hat{S}\}$ contains all the LLM-written segments.

One baseline approach could be to classify each $x_i$ individually using one of the binary classification approaches. However, these methods are designed for longer texts, and are known to perform poorly on short texts (Verma et al., 2024). Our results in Sec. 4 demonstrate the poor performance of this approach. Instead, we propose to "stitch" together different segments of $x$ to construct stitched texts, as shown on the left side of Fig. 6. Concretely, let $S \subseteq \{1, \ldots, n\}$ be an index set which we use to select a subset of the segments in $x$. Define $x[S]$ to be the text formed by concatenating the segments of $x$ indexed by $S$ in the order of the indices. For example, if $S = \{1, 3, 4\}$, then $x[S] = x_1 x_3 x_4$. These stitched texts $x[S]$ mostly consist of multiple segments, and should be of sufficient length for some of the binary classifier methods to be effective. Drawing from observations in Fig. 3, suppose that $S$ selects all the LLM segments, and $S^\complement := \{1, \ldots, n\} \setminus S$ selects all the human segments. Then $x[S]$ will be fully-human, and $x[S^\complement]$ fully-LLM, and this should result in the largest discrepancy between $T(x[S])$ and $T(x[S^\complement])$. On the other hand, if $x[S]$ contains a mixture of LLM and human text, then so will $x[S^\complement]$, and their $T$-scores should be more similar.

Thus, our goal is to find the $S$ that the $T$-score maximally discriminates $x[S]$ and $x[S^\complement]$:

$$
\hat{S} = \underset{S \subseteq \{1, \ldots, n\}}{\arg\max} \ f_x(S) := T(x[S]) - T(x[S^\complement]).
\tag{4}
$$

The overall method, which we call PaLD-TI, is shown in Fig. 6. In practice, we disregard $S = \emptyset$ and $S^\complement = \emptyset$, yielding a total of $2^n - 2$ sets to consider. This is because those two cases reduce the LLM text identification problem to whole text classification, which has been well explored in the previous literature. Thus, Eq. 4 is a subset selection problem which is combinatorial and has complexity exponential in $n$. In our experiments, we consider texts with at most $n = 10$ segments which is feasible to solve Eq. 4 exactly. For texts with more

---

**Algorithm 1** Greedy algorithm, PaLD-TI.

Initialize $S = \{\arg\max_{e \in \{1,\ldots,n\}} f_x(\{e\})\}$

Initialize $A = \{1, \ldots, n\} \setminus S$

**while** $f_x(S)$ increases **do**

$\quad e' = \arg\max_{e \in A} f_x(S \cup e) - f_x(S)$

$\quad S \leftarrow S \cup e$

$\quad A \leftarrow A \setminus e$

**end while**

---

sentences, one can chunk the text into paragraphs before solving Eq. 4, or use approximate algorithms (e.g., greedy) that trade-off optimality for efficiency. For example, the greedy algorithm will iteratively build a set by adding a single segment that maximizes the marginal gain of $f_x(S)$ at each iteration; see Alg. 1. This reduces the complexity to $O(n^2)$ but solves Eq. 4 approximately rather than exactly. We also note that PaLD-TI can be used for percentage estimation via the total predicted LLM length.

## 4 EMPIRICAL STUDY

**Datasets.** We evaluate our methods on the WritingPrompts (WP) (Fan et al., 2018) and Yelp Reviews (Yelp) (Yelp, 2014) datasets which are typically used to benchmark LLM text detection. WP contains pairs of story prompts and human-written short stories as responses, and Yelp contains human-written reviews of various businesses. Both datasets come with the fully-human and fully-LLM (GPT3-rewritten) versions of text; in additional to these, we adopt a sentence-level mask-and-fill approach with GPT-4o (Achiam et al., 2023) to generate mixed texts at different LLM fractions for both datasets, described in full in Sec. A.1. In total, for each dataset, we generate 3,600 and 300 mixed texts for training and test splits, respectively. For the training split, the LLM target fractions are ranged from $0.1, 0.2, \ldots,$ to $0.9$; while the LLM target fractions are set to $0.25, 0.5, 0.75$ for the test split, and the amount of data at each fraction are similar as a balanced dataset. Note that we measure the LLM fraction at character-level, and the training and testing splits do have a similar distribution of LLM fractions over $[0, 1]$ (see Tab. 9).

**Experimental Setup.** For the $T$-score used in both PaLD-PE and PaLD-TI, we adopt the logits of a RoBERTa classifier trained on fully-human and fully-LLM text, using the LogitNorm loss (Wei et al., 2022), as discussed in Sec. 3. Further hyperparameter choices for PaLD are described in Sec. A.2. For the percentage estimation (point estimate) and LLM text identification baselines, we adopt the existing LLM text detectors applied segment-wisely to the mixed-text samples, including DetectGPT-Seg (Mitchell et al., 2023), FastDetectGPT-Seg (Bao et al., 2024), Ghostbuster-Seg (Verma et al., 2024). DetectGPT and FastDetectGPT's threshold is set to maximize the difference between true-positive and false-positive rates when classifying human/LLM segments. In addition, we build another two baselines by fine-tuning the RoBERTa (Liu et al., 2019) as a binary classifier on fully-human and full-LLM texts with two different losses. One is trained with cross-entropy loss (RoBERTa) and another one is trained with LogitNorm (RoBERTa-LN). The predicted percentage is then the total character length of the predicted model segments divided by the total number of characters of the text. We use PaLD-TI here in a similar fashion. Moreover, given the generated mixed-text data and LLM fractions, we include another two baselines by fine-tuning the RoBERTa model with regression loss (RoBERTa-Reg) and squared loss and quantile loss (RoBERTa-QuantileReg)(Padilla et al., 2022; Koenker & Bassett, 1978). For RoBERTa-QuantileReg, we use the $\frac{\alpha}{2}$, $\frac{1}{2}$, and $1 - \frac{\alpha}{2}$ quantiles, with the $\frac{1}{2}$-quantile serving as a point estimate and the other two serving as a $1 - \alpha$ interval estimate.

**Evaluation.** We report performance averaged across the data with LLM fractions $\delta = 0.25, 0.5, 0.75$. For the percentage estimation task, we use mean absolute error (MAE), $|\hat{\delta} - \delta|$ to evaluate the point estimates; for interval estimates we report two metrics: *coverage* (C), the frequency of the interval covers $\delta$ (i.e., $\mathbb{E}_{X,\delta}[\mathbb{1}\{\widehat{\delta_L} \leq \delta \leq \widehat{\delta_R}\}]$, where $X$ are mixed texts with LLM fraction $\delta$ drawn from our dataset), and *precision* (P), the width of the interval (i.e., $\mathbb{E}_X[\widehat{\delta_R} - \widehat{\delta_L}]$). In general, one cannot always increase coverage and minimize precision, as a smaller interval reduces the probability that $\widehat{\delta_L} \leq \delta \leq \widehat{\delta_R}$; we report the trade-off between these two quantities. For the text identification task, we report segment-wise accuracy (i.e., $\mathbb{E}_X[\frac{1}{n}\sum_{i=1}^n \mathbb{1}\{i \in S, \hat{S} \quad \text{or} \quad i \in S^\complement, \hat{S}^\complement\}]$, where $S$ contains

Table 2: Percentage estimation results (point estimate); mean absolute error. The best method is bold.

| Dataset | PaLD-PE (Ours) | PaLD-TI (Ours) | RoBERTa-Reg | RoBERTa-QuantileReg | DetectGPT-Seg | FastDetectGPT-Seg | RoBERTa-Seg | RoBERTa-LN-Seg | Ghostbuster-Seg |
|---|---|---|---|---|---|---|---|---|---|
| WP | **0.116** | 0.122 | 0.207 | 0.186 | 0.370 | 0.184 | 0.342 | 0.434 | 0.215 |
| Yelp | **0.137** | 0.158 | 0.181 | 0.175 | 0.397 | 0.190 | 0.382 | 0.437 | 0.282 |

Table 3: LLM text identification results; segment-wise and top-1 accuracy. The best method is bold.

| Dataset | PaLD-TI (Ours) | | PaLD-TI-Greedy (Ours) | | DetectGPT-Seg | | FastDetectGPT-Seg | | RoBERTa-Seg | | RoBERTa-LN-Seg | | Ghostbuster-Seg | |
|---|---|---|---|---|---|---|---|---|---|---|---|---|---|---|
| | Seg | Top-1 | Seg | Top-1 | Seg | Top-1 | Seg | Top-1 | Seg | Top-1 | Seg | Top-1 | Seg | Top-1 |
| WP | **0.826** | **0.252** | 0.741 | 0.071 | 0.486 | 0.001 | 0.651 | 0.063 | 0.635 | 0.070 | 0.502 | 0.000 | 0.599 | 0.053 |
| Yelp | **0.801** | **0.247** | 0.733 | 0.069 | 0.452 | 0.002 | 0.624 | 0.054 | 0.560 | 0.013 | 0.506 | 0.000 | 0.540 | 0.027 |

ground-truth indices of segments of $X$ that were LLM-written), which measures how well each segment is classified. In addition, we report the top-1 accuracy at the set level (i.e., $\mathbb{E}_X[\mathbb{1}\{S = \hat{S}\}]$), which measures how often the ground-truth set $S$ is perfectly recovered, of which there are $2^n - 2$ possible for each mixed text. To achieve high top-1 accuracy is challenging, as a success requires *every* segment of the text to be correctly identified.

## 4.1 IN-DOMAIN MIXED-TEXT DETECTION

**Results on Percentage Estimation.** Tab. 2 compares our methods (PaLD-PE and PaLD-TI) with other methods on the point estimate results. PaLD-PE significantly outperforms the baselines that adopt a segment-wise strategy, demonstrating the limitations of existing LLM text detectors, likely due to the shortness of the individual segments. On the other hand, RoBERTa-Reg and RoBERTa-QuantileReg are the most competitive baselines while our method still surpasses them, showing that the Bayesian approach provides more precise estimates than direct regression. PaLD-TI is superior to all baselines, but not as accurate as PaLD-PE, since its precision is limited to segment-level.

For interval predictions, we sweep $\alpha$'s for both PaLD-PE (corresponding to the $(1 - \alpha)$-HDI) and RoBERTa-QuantileReg (corresponding to the $\frac{\alpha}{2}$ and $1 - \frac{\alpha}{2}$ quantiles) to get a coverage-precision trade-off. Fig. 7a shows our method yield a superior coverage-precision trade-off compared to RoBERTa-QuantileReg. For example, on WP, under similar coverage level, e.g., at around 77%, PaLD-PE produces tighter interval estimates than RoBERTa-QuantileReg (0.33 v.s. 0.60). We note that for both methods, the targeted coverage level of $1 - \alpha$ does not exactly match the achieved coverage level; but our method provides closer estimate, which allows the users to approximate the coverage before tuning $\alpha$. We attribute this to our model of $P(T(X)|\Delta)$ being an approximation of the underlying data generation process.

**Results on LLM Text Identification.** Tab. 3 shows the LLM text identification performance. We see that PaLD-TI outperforms the segment-wise baselines by at least 18% in terms of segment-wise accuracy. We also implement the approximate version of PaLD-TI by solving Eq. 4 using the greedy algorithm in Alg. 1. As expected, it does not perform as well as solving Eq. 4 exactly, but still outperforms the other baselines by at least 9% in

Table 4: PaLD-TI top-$p$ performance.

| Dataset | Top-0.05 | Top-0.20 |
|---|---|---|
| WP | 0.743 | 0.876 |
| Yelp | 0.607 | 0.830 |

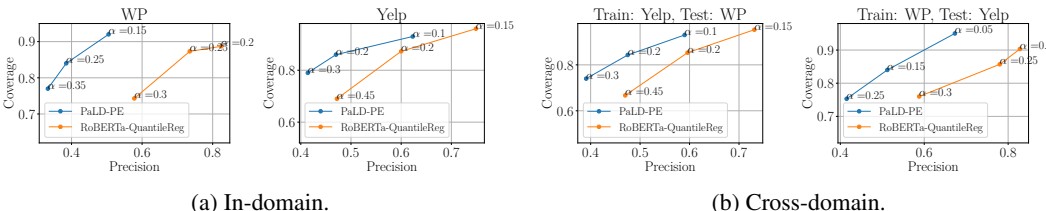

(a) In-domain.                    (b) Cross-domain.

Figure 7: Percentage estimation results, in terms of the coverage-precision trade-off. Upper-left indicates better performance; $\alpha$ sets the desired coverage level.

terms of segment-wise accuracy. On a single A10 GPU, the exact solver takes $\approx 30$s for a 10-segment text on average, whereas greedy takes $\approx 2.1$s. On the other hand, PaLD-TI outperforms all other baselines by over 20% in terms of top-1 accuracy, and no other baseline is able to achieve over 7% top-1 accuracy. To achieve high Top-1 accuracy is a difficult task as it requires all segments to be successfully identified. Moreover, as PaLD-TI can also rank the predictions, we also report the top-$p$ accuracy, which measures how often the ground-truth segments lie in the top $p$ fraction of all $f_x(S), S \subseteq \{1, \ldots, n\}$, in Tab. 4. We see that even if the ground-truth is not recovered by $\hat{S}$, it is in the top $0.05$ fraction of our objective function more than 60% of the time, and in the top $0.2$ fraction over 83% of the time. This further supports the validity of $f_x(S)$ in determining the LLM segments.

## 4.2 Cross-Domain Mixed-Text Detection

In this section, we evaluate PaLD when the training/tuning domain and testing domain does not match. For the PE and TI tasks, we compare with the two best baselines in Tabs. 2, 3, respectively.

**Fixed LLM.** Here, the LLM used to generated mixed-texts is fixed as GPT-4o. For PaLD-PE, the likelihood fitting phase is trained on WP, and the point estimate or predictive interval is estimated on the testing dataset of a different domain; similarly, the baselines are trained on WP and evaluated on a different testing domain. The out-of-domain datasets we evaluate on include Yelp (Yelp, 2014) and RAID (Dugan et al., 2024). We use the Ab-

Table 5: Cross-domain fixed-LLM results for percentage estimation (mean absolute error) and text identification (segment-wise accuracy). Training dataset: WP; LLM: GPT-4o.

| Test Dataset | Percentage Estimation (MAE ↓) | | | Text Identification (Seg-Acc. ↑) | | |
|---|---|---|---|---|---|---|
| | PaLD-PE | RoBERTa-Reg | RoBERTa-QuantileReg | PaLD-TI | FastDetect-GPT-Seg | RoBERTa-Seg |
| Yelp | **0.147** | 0.211 | 0.231 | **0.801** | 0.626 | 0.592 |
| Abstracts | **0.151** | 0.306 | 0.223 | **0.718** | 0.693 | 0.656 |
| Wiki | **0.224** | 0.313 | 0.238 | 0.623 | 0.636 | **0.735** |

stracts and Wiki domains of RAID, and generate interleaved mixed-texts from GPT-4o (Sec. A.1). Tab. 5 shows that PaLD-PE remains the best-performing method compared to the baselines. This demonstrates that PaLD-PE has a superior ability to generalize across domains, and that the $T$-scores used do also generalize across domains. For the interval estimates, we illustrate the cross-domain performance between WP and Yelp in Fig. 7b. Similar to the point estimates, we again see that PaLD-PE yields a superior coverage-precision tradeoff to RoBERTa-QuantileReg. Moreover, the $\alpha$ values set for the respective algorithms to control the width of the predicted interval results in achieved coverage values that more closely align with $1 - \alpha$.

For TI, PaLD-TI does not require training, but may use a $T$-score that is fitted to a dataset (e.g., RoBERTa-LN trained on WP). The FastDetectGPT baseline (applied segment-wise) has threshold set to maximize TPR $-$ FPR on WP, and RoBERTa (segment-wise) is trained on WP. We see that PaLD-TI is superior to baselines on Yelp and Abstracts but not Wiki, where RoBERTa-Seg achieves high segment-wise accuracy.

**Cross-LLM.** Here, we evaluate the case when additionally the LLM used to train/tune may not match the LLM used to generate mixed-texts at test time. In particular, we evaluated PaLD and baselines trained/fitted on WP (GPT-4o-written) with the testing domains set as WP[3], Yelp, and RAID (Abstracts, Wiki), where the testing domain mixed-texts are generated in the interleaved manner following Sec. A.1 using Claude-3.5-Sonnet (Anthropic, 2024). In addition, we evaluate RoFT (Dugan et al., 2020), which was designed to evaluate human-LLM boundary detection, so the mixed texts were

Table 6: Cross-domain cross-LLM results for percentage estimation (MAE) and text identification (segment-wise accuracy). Training dataset: WP (GPT-4o).

| | Test Dataset | Percentage Estimation (MAE ↓) | | | Text Identification (Seg-Acc. ↑) | | |
|---|---|---|---|---|---|---|---|
| | | PaLD-PE | RoBERTa-Reg | RoBERTa-QuantileReg | PaLD-TI | FastDetect-GPT-Seg | RoBERTa-Seg |
| Claude | WP | **0.114** | 0.184 | 0.138 | **0.751** | 0.531 | 0.675 |
| | Yelp | **0.128** | 0.193 | 0.144 | **0.731** | 0.628 | 0.654 |
| | Abstracts | **0.115** | 0.204 | 0.145 | **0.710** | 0.633 | 0.651 |
| | Wiki | **0.174** | 0.245 | 0.185 | 0.602 | 0.634 | **0.683** |
| GPT-2 | SS | 0.227 | 0.236 | **0.227** | 0.555 | **0.610** | 0.525 |
| | Recipes | **0.166** | 0.209 | 0.186 | **0.548** | 0.520 | 0.505 |
| | NYT | 0.236 | 0.246 | 0.239 | 0.477 | **0.613** | 0.544 |

generated by prompting GPT2-XL to complete $10-k$ sentences when prompted with $k$ human-written sentences; we use the Short Stories (SS), Recipes, and New York Times (NYT) domains.

---

[3]In Tab. 6, all rows are cross-LLM and cross-domain, except for WP which is only cross-LLM.

Shown in Tab. 6, on Claude, PaLD-PE outperforms all baselines despite the domain and LLM shift. PaLD-TI is superior on all domains except Wiki, where RoBERTa-Seg again achieves high accuracy. For GPT-2 (i.e., RoFT data), PaLD-PE performs similar or better. However, for the TI task, both PaLD-TI and RoBERTa-Seg result in a significant performance decrease compared to FastDetectGPT; this may be due to poor generalization to GPT-2 of the RoBERTa-based classifiers that is used for both RoBERTa-Seg and the $T$-score for PaLD-TI. The better generalization to Claude from GPT-4o compared to GPT-2 may be expected, as the parameter count of GPT-2 is orders of magnitude lower compared to Claude and GPT-4o.

### 4.3 PARAPHRASING ATTACKS

In the binary classification setting, recent work has shown that LLM detectors such as Detect-GPT are easily attacked by paraphrasing attacks (Sadasivan et al., 2023; Krishna et al., 2024).

Following (Sadasivan et al., 2023), we use the T5-based Parrot paraphraser (Damodaran, 2021) to randomly paraphrase the LLM-written segments of the mixed texts. We randomly paraphrase 25%, 50% and 100% of the LLM-written segments of the WP dataset, and evaluate the PE and TI tasks in Tab. 7. We observe that all methods suffer performance decreases

Table 7: Percentage estimation (MAE) and text identification (segment-wise accuracy) under paraphrase attacks on LLM-written segments in WP.

| Fraction Chosen | Percentage Estimation (MAE ↓) | | | Text Identification (Seg-Acc. ↑) | | |
|---|---|---|---|---|---|---|
| | **PaLD-PE** | RoBERTa-Reg | RoBERTa-QuantileReg | **PaLD-TI** | FastDetect-GPT-Seg | RoBERTa-Seg |
| 0.25 | **0.117** | 0.204 | 0.193 | **0.811** | 0.633 | 0.701 |
| 0.50 | **0.122** | 0.204 | 0.193 | **0.786** | 0.624 | 0.691 |
| 1.00 | **0.134** | 0.207 | 0.195 | **0.730** | 0.584 | 0.679 |

with increasing attack strength. However, even when all LLM segments are attacked, PaLD still maintains superior performance compared to baselines. We leave further analysis of mixed-text detector attacks for future work.

### 4.4 ABLATION STUDY

We investigate several critical components of PaLD on the WP dataset, namely $T$-score. In Appendix A.3, we further show how in PaLD-PE modelling $\phi_k$ in Eq. 2 as a KDE is superior to simpler assumptions such as Gaussian (Tab. 14), and the effect of choosing the Beta prior's parameters (Tab. 15) and the KDE bandwidth (Tab. 13). We also show the effect of baselines finetuned on segment-level text (Tab. 10, 11, 19, 20), the $F_1$-scores in text identification (Tab. 12), the effect of segmentation mismatch (Tab. 18), and how PaLD performs on all- or no- LLM segments (Tab. 16, 17).

$T$-**score.** In Sec. 3, we analyze the characteristics of different $T$-scores by $\sigma_T^*$ in Tab. 1, here we further evaluate them on both percentage estimation and LLM text identification on the WP dataset. Shown in Tab. 8, we see that RoBERTa-LN yields the best performance for both percentage estimation and LLM text identification.

Table 8: $T$-score ablation on the WP dataset.

| T-score | $\sigma_T^*$ | Percentage Estimation | | | Text Identification | |
|---|---|---|---|---|---|---|
| | | MAE | C↑ | P↓ | Top-1 Acc | Seg Acc |
| RoBERTa-LN | 2.599 | 0.116 | 84% | 0.385 | 0.252 | 0.826 |
| RoBERTa | 2.160 | 0.163 | 86% | 0.526 | 0.205 | 0.798 |
| Ghostbuster | 1.889 | 0.151 | 86% | 0.553 | 0.044 | 0.619 |
| FastDetectGPT | 1.784 | 0.235 | 85% | 0.671 | 0.110 | 0.653 |
| DetectGPT | 1.496 | 0.169 | 84% | 0.547 | 0.101 | 0.690 |

Moreover, the performance of both tasks correlates with the magnitude of the average quantile slope $\sigma_T^*$ defined in Eq. 1. This lends support to our observation that a larger $\sigma_T^*$ indicates increased separation of the $T$-score distribution as it varies with $\delta$. This provides increased statistical power to predict $\delta$ from a text $T$-score, and yields improved precision of the predicted interval. Similarly, in text identification, it provides a $f_x(S)$ that is more strongly indicative of LLM text, yielding better segment-accuracy and top-1 accuracy with increased $\sigma_T^*$.

## 5 FINAL REMARKS

PaLD-TI is NP-hard and does not scale well with the number of segments. While we demonstrate the greedy algorithm as a low-complexity solution that can scale with many segments, better performance at low complexity may be desirable, which we leave for future work. Additionally, PaLD-TI requires a fixed segmentation. A chosen segmentation may not align perfectly with ground-truth segmentation. Future work can investigate the effect of segment misalignment, extend to hallucination detection, and settings such as humans mimicking LLM styles.

**Disclaimer.** This paper was prepared for informational purposes by the Global Technology Applied Research center of JPMorgan Chase & Co. This paper is not a product of the Research Department of JPMorgan Chase & Co. or its affiliates. Neither JPMorgan Chase & Co. nor any of its affiliates makes any explicit or implied representation or warranty and none of them accept any liability in connection with this paper, including, without limitation, with respect to the completeness, accuracy, or reliability of the information contained herein and the potential legal, compliance, tax, or accounting effects thereof. This document is not intended as investment research or investment advice, or as a recommendation, offer, or solicitation for the purchase or sale of any security, financial instrument, financial product or service, or to be used in any way for evaluating the merits of participating in any transaction.

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

# A APPENDIX

## A.1 MIXED TEXT GENERATION

We use a mask-and-fill approach to generate synthetic mixed texts. This process is used for both generating the histograms in Fig. 3 and fitting the mixture KDE model in Sec. 3. Specifically, if we denote $x^H$ as a text from our human-written dataset (e.g. WritingPrompts or Yelp Reviews), then $x^H = x_1^H \dots x_p^H$ is segmented at the sentence level. Then, segments are randomly masked, to form a masked text $x^M$. For example, a 5-segment human text $x^H = x_1^H x_2^H x_3^H x_4^H x_5^H$, masked at sentences 2 and 5, would become $x^M = x_1^H$ [MASK] $x_3^H x_4^H$ [MASK]. The LLM is then prompted with the following prepended to $x^M$:

"What sentences should go in the $n_{\text{replace}}$ [MASK] locations of the following text? Only provide exactly one sentence per [MASK] location. Only provide the sentences as a numbered list with $n_{\text{replace}}$ sentences total."

Here, $n_{\text{replace}}$ is the number of [MASK] symbols in $x^M$. We found that when prompted in this way, GPT-4o successfully returns a numbered list containing exactly $n_{\text{replace}}$ sentences corresponding to the [MASK] symbols, with a failure rate of 1-2%. When successful, the sentences in the numbered list are inserted into the [MASK] positions of the masked text, to form the mixed text. We show a few examples of these mixed texts in Tab. 21.

Since this process does not control the length of the segments returned by GPT-4o, it is not easy to exactly control the fraction of LLM text $\delta$ at the character level. As a result, we target $\delta$ approximately by masking and filling a $\delta$ fraction of the human sentences. This is done for both the training split (targeting $\delta$ approximately at $0.1, 0.2, \dots, 0.9$) and testing splits (targeting $\delta$ approximately at $0.25, 0.5, 0.75$). The true $\delta$ (at the character level), which can be computed post-hoc, is used for (i) the binning step in PaLD-PE collect the $T$-score samples for the $P(T(X)|\Delta = \delta_k)$ distributions, and (ii) to benchmark the performance metrics (mean absolute error, coverage, and prediction) for percentage estimation.

We apply this method using both GPT-4o and Claude-3.5-Sonnet (Anthropic, 2024).

**A note on the training and testing split.** Note that the *target* fractions for the testing split are 0.25, 0.5, and 0.75, but as described above, these are not actually the *true* LLM text percentages measured at the character-level, but rather the target fractions, which determine the number of masked sentences for the mask-and-fill procedure. Approximately 0.25, 0.5, and 0.75 of the sentences are masked out, then the true LLM fraction $\delta$ values are computed post-hoc. The true fractions depend on the text content and thus vary. We found that this procedure yielded similar distributions of $\delta$ over $[0, 1]$ compared to setting the target fractions to 0.1, 0.2, ..., 0.9 as done for the training split. To provide concrete evidence for this, we compute a histogram of the $\delta$ values on the Yelp dataset for both train and test splits in Tab. 9. Thus, despite the "target" fractions differing between the train and test split,

Table 9: Histogram of $\delta$ values in $[0, 1]$ for train and test splits of Yelp.

| Split | [0,0.1] | [0.1,0.2] | [0.2,0.3] | [0.3,0.4] | [0.4,0.5] | [0.5,0.6] | [0.6,0.7] | [0.7,0.8] | [0.8,0.9] | [0.9,1.0] |
|-------|---------|-----------|-----------|-----------|-----------|-----------|-----------|-----------|-----------|-----------|
| Train | 0.062 | 0.155 | 0.140 | 0.132 | 0.0972 | 0.0988 | 0.0957 | 0.106 | 0.0819 | 0.0315 |
| Test | 0.030 | 0.1367 | 0.170 | 0.1630 | 0.130 | 0.090 | 0.107 | 0.110 | 0.05330 | 0.01 |

the true fractions mostly match.

## A.2 PaLD IMPLEMENTATION DETAILS

During the distribution-fitting stage of PaLD-PE, the entire training split of a dataset is masked and filled using the above procedure, where the fraction of sentences masked is approximately $\delta_1 = 0.1, \delta_2 = 0.2, \dots, \delta_9 = 0.9$. We then we compute $T$-scores across all the text samples using the logits of a RoBERTa model trained with the LogitNorm loss Wei et al. (2022). We found this $T$-score to work the best for percentage estimation, as the LogitNorm improves calibration of the model which is necessary for the $T$-score to smoothly interpolate between fully-human and fully-LLM. We use LogitNorm with temperature $\tau = 0.005$, and train the RoBERTa model on the training split for

Table 10: Percentage estimation (MAE); RoBERTa classifiers trained on segment-level data.

| Dataset | RoBERTa-Seg | RoBERTa-LN-Seg |
|---------|-------------|----------------|
| WP | 0.566 | 0.434 |
| Yelp | 0.563 | 0.563 |

Table 11: LLM text identification (segment-wise accuracy); RoBERTa classifiers trained on segment-level data.

| Dataset | RoBERTa-Seg | RoBERTa-LN-Seg |
|---------|-------------|----------------|
| WP | 0.498 | 0.502 |
| Yelp | 0.496 | 0.496 |

Table 12: LLM text identification (segment-wise $F_1$-score).

| Dataset | PaLD-TI | FastDetectGPT-Seg |
|---------|---------|-------------------|
| WP | 0.798 | 0.480 |
| Yelp | 0.776 | 0.476 |

Table 13: Effect of KDE bandwidth.

| Bandwidth $h$ | PaLD-PE (MAE) |
|---------------|---------------|
| 0.01 | 0.129 |
| 0.1 | 0.118 |
| 1 | 0.124 |
| $0.312 \pm 0.025$ (Scott) | 0.116 |

the respective datasets. These $T$-scores and then binned to $\delta = 0.125, 0.25, 0.375, \ldots, 0.875$, using the character-level fraction of LLM text in the mixed text. Then, a KDE with Gaussian kernel, with bandwidth chosen using Scott's rule Scott (1992), is fit to the $T$-scores at each level of $\delta_k$ to form the likelihood model $P(T|\delta)$ as described in the main text. For the posterior, we choose the prior $P(\delta)$ to be the Beta$(2, 2)$ distribution. During the inference stage, we sample 5000 samples, discarding the first 1000 due to burn-in, using Metropolis-Hastings (Gelman et al., 2004) with a proposal distribution as the truncated normal centered at the previous sample, truncated to $[0, 1]$. The predicted percentage and interval are given by the MAP estimate and $(1 - \alpha)$-HDI interval, respectively.

### A.3 ADDITIONAL RESULTS AND ABLATIONS

**Baseline RoBERTa classifiers applied to segments.** The baseline RoBERTa classifiers applied segment-wise, for both the PE and TI tasks, were trained on paragraph-level fully human or fully-LLM texts. As this may incur a distribution shift when applied to segments, we train the RoBERTa classifiers to sentence-length data gathered from the mixed-text training splits (Sec. A.1).

Similar to Sec. 4, for PE, these baselines return the ratio of predicted LLM characters to total characters, and for TI, classify each segment individually. Shown in Tabs. 10, 11, the performance is significantly worse than the RoBERTa classifiers trained on paragraph-level texts, reported in Tab. 2 and Tab. 3 for PE and TI. This is likely due to the texts being too short, as RoBERTa is pretrained on longer texts, and short texts do not offer enough context, making it difficult to extract statistical dependence with its origin.

**$F_1$-score of LLM text identification.** To measure the impacts of both the false-negative rate (FNR) and false-positive rate (FPR) of the LLM text identification task, we further report the segment-wise F1-score for the text identification task in Tab. 12. We also report the F1-score of FastDetectGPT applied segment-wise, which was the strongest baseline in terms of accuracy in Tab. 3. As shown, PaLD-TI significantly outperforms it in terms of F1-score as well, demonstrating a good balance between precision and recall compared to baselines. In summary, the PaLD-TI outperforms FastDetectGPT not only in its accuracy (Tab. 3), but also in the F1-score (Tab. 12).

**Distributional assumptions for PaLD-PE.** For PaLD-PE, we choose a mixture KDE for the likelihood and a Beta distribution for the prior. Here, we analyze how the choice of how we model the likelihood affects the performance of PaLD-PE. We compare the mixture KDE with a mixture Gaussian and mixture Cauchy. Namely, we fit $\phi_k(t)$ in Eq. 2, to the samples $\{t_i^{(k)}\}_{i=1}^{n_k}$ using Gaussian and Cauchy distributions with RoBERTa-LN $T$-scores. Shown in Tab. 14, using a KDE performs significantly better, demonstrating that the higher expressivity of the distribution has a large influence on percentage estimation performance. We also varied the $a, b$ parameters of the Beta$(a, b)$ prior

Table 14: Likelihood model ablation on the WP dataset.

| $\phi_k(t)$ | Percentage Estimation | | |
|---|---|---|---|
| | MAE | C $\uparrow$ | P $\downarrow$ |
| KDE | 0.116 | 84% | 0.385 |
| Normal | 0.200 | 85% | 0.630 |
| Cauchy | 0.227 | 85% | 0.710 |

Table 15: $\text{Beta}(a, b)$ prior ablation on the WP dataset with KDE likelihood model.

| $a, b$ | Percentage Estimation | | |
| | MAE | C $\uparrow$ | P $\downarrow$ |
|---|---|---|---|
| 1, 1 | 0.155 | 84% | 0.450 |
| 2, 2 | 0.116 | 84% | 0.385 |
| 3, 3 | 0.122 | 84% | 0.395 |

Table 16: Percentage estimation on all or no LLM segments; mean absolute error.

| Dataset | PaLD-PE | FastDetectGPT |
|---|---|---|
| No LLM | 0.137 | 0.261 |
| All LLM | 0.220 | 0.366 |

Table 17: Text identification on all or no LLM segments; segment accuracy.

| Dataset | PaLD-TI | FastDetectGPT |
|---|---|---|
| No LLM | 0.900 | 0.829 |
| All LLM | 0.728 | 0.718 |

between $1 \leq a, b \leq 3$ in Tab. 15, and $a = b = 2$ perform best, but this may be dependent on the distribution of $\delta$ encountered at test time.

**Ablation study on KDE bandwidth selection.** Below, we report the effect of the bandwidth ($h$) on PaLD-PE's performance on the WP dataset. As mentioned in the appendix, we use Scott's rule to automatically set the bandwidth for results in the main text. In Tab. 13, we additionally report the average bandwidths chosen by Scott's rule over all the $\phi_k$'s in Eq. 2, with one standard deviation. As shown, PaLD-PE is not too sensitive to the choice of bandwidth, and Scott's rule is able to determine a bandwidth resulting in accurate estimates of the LLM fraction.

**All or No LLM Segments** As the setting of this paper aims for mixed-text, we did not include the No LLM and the All LLM cases, as these two cases reduce to the classic setting of existing LLM text detection frameworks such as (Fast)DetectGPT and Ghostbuster. On the WP dataset, we evaluate the PaLD framework on the No LLM and the All LLM (GPT-4o) cases (Tabs. 16, 17) and compare it with the FastDetectGPT, which outperforms DetectGPT and Ghostbuster in Tab 2. Note that FastDetectGPT is initially designed for the binary classification between No LLM and the All LLM.

The results further suggest that even on the classic setting, PaLD still provides competitive estimations. Note that PaLD-TI cannot pick all segments to be LLM or not LLM, and therefore only considers $2^n - 2$ of the $2^n$ possibilities. Thus, it will always make at least one mistake. This means the 0.900 segment accuracy for No LLM segments is the best it can achieve, since there are 10 segments per text in this dataset, and PaLD-TI will misclassify at least one segment incorrectly. Additionally, we would like to reiterate our response to Q3 that the testing datasets contain texts that are nearly all LLM or no LLM segments.

**Effect of segmentation mismatch in PaLD-TI.** PaLD-TI, as well as the segment-wise baselines, assumes a chosen segmentation, which we assume to be sentences. Here, we investigate the effect of a mismatched segmentation. In other words, the case when the ground-truth segmentation (i.e., the actual segments where the text is fully LLM or fully human) does not perfectly align with the chosen segmentation of PaLD-TI. To do so, we use the same sentence-level mixed text, so the ground-truth segmentation is at the sentence-level. However, the chosen segmentation is now at every 2 sentences. This means that PaLD-TI (and segment baselines) can only assign prediction to pairs of sentences, and thus a misclassification (at the sentence-level) will occur whenever the two sentence within a pair differ in class (i.e., one is human and the other is LLM), no matter what the prediction is. We show how PaLD-TI and the FastDetectGPT-Seg perform in Tab 18.

As shown, there is a performance decrease in both PaLD-TI and FastDetectGPT-Seg, as expected, since perfect classification is no longer possible and mistakes are inevitable. We note that PaLD-TI still outperforms FastDetectGPT-Seg at the 2-sentence segmentation, and leave methods to mitigate mismatched segmentation for future work.

Table 18: Text identification under mismatched segmentation. Dataset: WP. Ground-truth segmentation is at the sentence-level. Report sentence-wise accuracy.

| Chosen Segmentation | PaLD-TI | FastDetectGPT-Seg |
|---|---|---|
| Every Sentence | 0.826 | 0.651 |
| Every 2 Sentences | 0.706 | 0.613 |

Table 19: Percentage estimation (MAE $\downarrow$)

| Dataset | DetectGPT (para.) | DetectGPT (sent.) | FastDetectGPT (para.) | FastDetectGPT (sent.) | PaLD-PE (ours) |
|---|---|---|---|---|---|
| WP | 0.381 | 0.370 | 0.212 | 0.184 | 0.116 |
| Yelp | 0.407 | 0.397 | 0.218 | 0.190 | 0.137 |

Table 20: Text identification (segment accuracy $\uparrow$)

| Dataset | DetectGPT (para.) | DetectGPT (sent.) | FastDetectGPT (para.) | FastDetectGPT (sent.) | PaLD-TI (ours) |
|---|---|---|---|---|---|
| WP | 0.486 | 0.488 | 0.650 | 0.651 | 0.826 |
| Yelp | 0.452 | 0.460 | 0.622 | 0.624 | 0.801 |

**Baseline DetectGPT and FastDetectGPT methods applied to segments.** We compare the DetectGPT and FastDetectGPT segment-wise baselines by setting the threshold to maximize the difference between TPR and FPR on a segment-level validation set, versus on a paragrpah-level dataset. We show the difference by setting threshold at the paragraph level vs. sentence level in Tab. 19, 20. There is an improvement in the performance by using sentence-level validation data, albeit around $< 0.03$ in MAE for the percentage estimation task, and $< 1\%$ for the segment-accuracy in the text identification task.

Table 21: Examples of synthetic mixed text generation using the mask-and-fill approach with GPT-4o. Text highlighted in red were originally segments masked in the human text and filled by GPT-4o.

| Human Text | Mixed Text |
|---|---|
| Thank you for a lovely morning!  I was in NJ early and decided to stop in for a delicious diner breakfast.  I got a taylor ham, egg, and cheese and a short stack of blueberry pancakes (I simply couldn't decide between sweet or savory, plus leftovers are never a bad thing).  The egg sandwich was fantastic. I was nervous when they said they didn't have kaiser rolls so I went with a hamburger roll. It was an excellent decision.  The roll was excellent – not just an average cheap hamburger roll.  They layered the cheese on the sandwich, which to me is a must for a true egg sandwich.  It was served with home fries, which were sauteed with large slices of peppers and onions. | *This morning, I decided to treat myself to breakfast at a local diner.*  I got a taylor ham, egg, and cheese and a short stack of blueberry pancakes (I simply couldn't decide between sweet or savory, plus leftovers are never a bad thing).  The egg sandwich was fantastic.  I was nervous when they said they didn't have kaiser rolls so I went with a hamburger roll.  It was an excellent decision. *The pancakes were fluffy and bursting with blueberries.*  They layered the cheese on the sandwich, which to me is a must for a true egg sandwich.  *The meal also came with a side of breakfast potatoes.* |
| This was a very busy place.  Was told I had to try this place while I was in St Louis.  I was not disappointed.  I got a peach shake that was amazing.  The rest of my group tried a number of different options that they all enjoyed.  The wait was a little long especially on a very hot date.  The prices were very reasonable.  The ordering at the window was confusing.  Multiple windows with not a lot of direction of which line to get in and where to wait for the food.  It's a great place to stop if you are in the area. | *I recently visited a local ice cream shop. The place had a charming, old-fashioned vibe.* I was not disappointed.  I got a peach shake that was amazing.  *The staff was friendly and helpful.  The menu had a wide variety of flavors and treats.*  The prices were very reasonable.  The ordering at the window was confusing.  Multiple windows with not a lot of direction of which line to get in and where to wait for the food.  It's a great place to stop if you are in the area. |
| A man is banished to the wilderness for 20 years.  Write his diary entries for his first and last days of exile.  I was born to fire. It flowed over my skin, danced upon my face, and stripped me of what little humanity I had left.  Within the ruined cavity of my left eye I held the final images of my family as they were fed to the same fires I was pulled from.  My death would not be so quick and so I was allowed to burn with them, but live. As soon as I was able to walk, I was ushered out into the wilderness.  The final piece of society I was allowed to keep was in the ink buried in my chest that had once formed my son's hand print, now twisted with my burned skin into a misshapen claw.  They promised twenty years, but swore under their breath | A man is banished to the wilderness for 20 years.  Write his diary entries for his first and last days of exile.  *Today marks the beginning of my exile, a punishment I must endure for the next two decades.  The pain of separation from my loved ones is unbearable, but I must find the strength to survive.* Within the ruined cavity of my left eye I held the final images of my family as they were fed to the same fires I was pulled from. My death would not be so quick and so I was allowed to burn with them, but live.  As soon as I was able to walk, I was ushered out into the wilderness.  *The years have been long and arduous, but I have learned to find solace in the solitude of the wilderness.  As I take my final steps back to civilization, I carry with me the scars and wisdom of my exile* |
| Describe an object within five feet of you in as much detail as possible.  A pair of simple black converse lie on the floor of the baby blue Honda fit my girlfriend is kind enough to let me drive.  They are a far cry from the crisp kicks I'd received in the mail only a year ago.  This has been a hard 12 months for them.  The once crisp white inner lining has degraded into something a generous person might call "cream" or "off-white" to me they're just brown.  The forces of time have transmuted the laces into a soft grey, like clouds in fall which promise a gentle patter of rain to listen to as you while away the hours.  The rubber has had it particularly bad, time and constant use has worn down the bottom edges.  Scuff marks cover the once pristine expanse.  When they were new I'd taken, so much care to keep them scuff free I waited until the wedding to wear them. | Describe an object within five feet of you in as much detail as possible.  A pair of simple black converse lie on the floor of the baby blue Honda fit my girlfriend is kind enough to let me drive.  They are a far cry from the crisp kicks I'd received in the mail only a year ago.  *The laces are frayed and stained, no longer the bright white they once were.*  The once crisp white inner lining has degraded into something a generous person might call "cream" or "off-white" to me they're just brown.  *The rubber soles are worn down, evidence of countless steps taken. The black canvas is faded, showing signs of wear and tear from daily use.*  Scuff marks cover the once pristine expanse.  When they were new I'd taken, so much care to keep them scuff free I waited until the wedding to wear them. |

