# OpenReview forum: "PaLD: Detection of Text Partially Written by Large Language Models"
_ICLR.cc/2025/Conference — ICLR 2025 Poster_

### Official Review · Reviewer_qnTJ · 2024-10-29

**Soundness:** 3
**Presentation:** 3
**Contribution:** 3
**Rating:** 6
**Confidence:** 3

**Summary:**

This paper explores methods for detecting human and LLM-generated text in a mixed-text environment. A detector called PaLD is proposed, which is used for estimating the proportion of LLM-generated content in a text and determining which passages are written by LLMs.

**Strengths:**

The paper tackles the mixed text problem through the PaLD method, which is an innovation in current LLM detection techniques. Above all, this paper has a complete structure and provides relatively comprehensive experiments.

**Weaknesses:**

1. The effectiveness of this method seems contingent upon accurate segmentation of text into LLM and human-written parts. It would be helpful to explore how it performs under less ideal segmentation conditions or how it handles errors in segmentation.

2. The article mentions the performance of cross-domain detection, but just two types are involved here, which seems to be insufficient to validate the generalization ability of the model.

**Questions:**

1. The paper used quantitative T-score and KDE to analyze or differentiate text. It is suggested that the authors provide more detailed basis for parameter setting, such as the choice of bandwidth for KDE and how to determine the threshold of T-score.

2.  Authors provide performance metrics of the algorithm such as accuracy and coverage-accuracy trade-offs, but lacks the analysis of misclassifications (false positives and false negatives). It is recommended to explore or discuss in detail the possible causes of misclassification.

---

> ### Author Response · Authors · 2024-11-21
>
> We appreciate the reviewer's comments and recognizing the novelty of the mixed-text LLM detection setting. We hope that the following responses help resolve some of the concerns of the reviewer.
>
> **Response to weaknesses**:
>
> **W1**. This paper assumes that the segmentation occurs at the sentence-level, i.e., a minimal segment would be a single semantically meaningful sentence. This is a natural choice to make, as sentences are a natural way to segment text in human languages, and are thus a likely method humans may edit LLM written texts. It is also natural to use a sentence as the minimal unit of which a piece of text may be considered all LLM or all human. Otherwise, it may be extremely difficult to define what unit of text is considered all LLM or all human. Even at the sentence-level, an "all LLM" text may share words with the "all human" counterpart. This should not mean that only the shared words are "human" and the words in between the shared words of the "all LLM" text are considered LLM-written. Nevertheless, PaLD-TI does assume a predefined segmentation, which means that if the segmentation is misaligned with the ground-truth, then PaLD-TI will not be able to achieve perfect accuracy at the character-level. We note that PaLD-PE (percentage estimation) is agnostic to any specific segmentation.
>
> **W2**. Regarding additional experiments, as mentioned in the Initial Common Response, we are currently evaluating several new domains from the RoFT and RAID datasets, and will add these results to the manuscript shortly.
>
> **Response to questions**:
>
> **Q1**. A threshold for the $T$-score is not necessary for PaLD; the only requirement is to compute $T$-scores. In PaLD-PE, predictions are made by sampling from the posterior of $P(T(X)|\Delta)$. In PaLD-TI, we simply use the $T$-score to compute our objective function. Regarding ablation studies, Sec. 4.3 explores different $T$-scores, and different distributions to model the $P(T(X)|\Delta)$ conditional distribution for PaLD-PE. As you suggested, we will additionally add an ablation study on the choice of bandwidth for when the KDE is used to model this conditional distribution. Below, we report the effect of the bandwidth ($h$) on PaLD-PE's performance on the WP dataset. As mentioned in the appendix, we use Scott's rule to automatically set the bandwidth. Below, we additionally report the average bandwidths chosen by Scott's rule over all the $\phi_k$'s in Sec. 3.2, with one standard deviation. As shown, PaLD-PE is not too sensitive to the choice of bandwidth, and Scott's rule is able to determine a bandwidth resulting in accurate estimates of the LLM fraction.
>
> MAE of point estimates from PaLD-PE on the WP dataset.
>
> | Bandwidth $h$ | PaLD-PE |
> | ------------- | ------- |
> | 0.01 | 0.129 |
> | 0.1 | 0.118 |
> | 1 | 0.124|
> | $0.312 \pm 0.025$ (Scott) | 0.116 |
>
> **Q2**. The reviewer's suggestion is great as it is argued in LLM detection that the False Positive Rate (FPR) must be kept low [1], and poses a greater risk than False Negative Rate (FNR).  To measure the impacts of both the FNR and FPR, we further report the segment-wise F1-score for the text identification task in the tables below.
> We also report the F1-score of FastDetectGPT applied segment-wise, which was the strongest baseline in terms of accuracy in Table 3.
> As shown, PaLD-TI significantly outperforms it in terms of F1-score as well, demonstrating a good balance between precision and recall compared to baselines. In summary, the PaLD-TI outperforms FastDetectGPT not only in its accuracy (Table 3), but also in the F1-score (Table below).
>
> LLM text identification
> (segment-wise F1-score).
>
> | Dataset | PaLD-TI | FastDetectGPT-Seg |
> |---------|---------|---------|
> | WP | 0.798  | 0.480 |
> | Yelp |  0.776 | 0.476 |
>
> We plan to update the manuscript with these results. Percentage estimation is a regression task, so misclassifications are not applicable.
>
> **References**
>
> [1] Krishna, K., Song, Y., Karpinska, M., Wieting, J., & Iyyer, M. (2024). Paraphrasing evades detectors of ai-generated text, but retrieval is an effective defense. Advances in Neural Information Processing Systems, 36.

---

> > ### Comment · Reviewer_qnTJ · 2024-11-25
> >
> > Thanks for the authors' response. After reviewing the comments, I will keep the original score.

---

> > > ### Author Response · Authors · 2024-11-25
> > >
> > > We again thank the reviewer for their feedback. As described in the Second Common Response, we have now updated the manuscript to reflect the results shown in our initial point-by-point reply. For those comments, please refer to the Appendix A.3 (Q1, Q2) and the "Other Experimental Additions" section of the Second Common Response. In addition, we have provided new results to address comments from W1 and W2. We hope that these additional experimental results help alleviate your concerns, and we would be grateful if the score could be reconsidered in light of these updated results. These are summarized below:
> > >
> > > **W1**. As mentioned in our previous point-by-point reply to you, our paper assumes segmentation occurs at the sentence-level, which we argue is a natural choice to make. Nevertheless, to provide some analysis on what happens when the segmentation is misaligned, we have added an experiment in Appendix A.3 (Table 18) to investigate the effect of a mismatched segmentation. In other words, the case when the ground-truth segmentation (i.e., the actual segments where the text is fully LLM or fully human) does not perfectly align with the chosen segmentation of PaLD-TI. To do so, we use the same sentence-level mixed text, so the ground-truth segmentation is at the sentence-level. However, the chosen segmentation is now at every 2 sentences. This means that PaLD-TI (and segment baselines) can only assign prediction to pairs of sentences, and thus a misclassification (at the sentence-level) will occur whenever the two sentence within a pair differ in class (i.e., one is human and the other is LLM), no matter what the prediction is. We show how PaLD-TI and the FastDetectGPT-Seg perform below.
> > >
> > > | Chosen Segmentation | PaLD-TI | FastDetectGPT-Seg |
> > > |---------------------|----------|-------------------|
> > > | Every Sentence | 0.826 | 0.651 |
> > > | Every 2 Sentences | 0.706 | 0.613 |
> > >
> > > As shown, there is a performance decrease in both PaLD-TI and FastDetectGPT-Seg, as expected, since perfect classification is no longer possible and mistakes are inevitable. We note that PaLD-TI still outperforms FastDetectGPT-Seg at the 2-sentence segmentation, and leave methods to mitigate mismatched segmentation for future work.
> > >
> > > **W2**. Regarding new datasets and new domains, we have evaluated RoFT and RAID in Sec. 4.2 (Table 5 and 6). This also contains new results on cross-LLM evaluation, where we evaluate our method on mixed-texts generated by Claude. Please refer to the Second Common Response for a summary of results.

---

> > > > ### Author Response · Authors · 2024-11-28
> > > >
> > > > Dear reviewer qnTJ,
> > > >
> > > > Thank you once again for your review of our work. As the discussion period draws to a close, we would like to inquire if you had the chance to look at the updated manuscripts and results (after your last reply), have any additional questions, or found our responses to be helpful.
> > > >
> > > > Please let us know! We appreciate your time and feedback.

---

> > > > > ### Comment · Reviewer_qnTJ · 2024-11-28
> > > > >
> > > > > Thank authors for your update. I have read the latest manuscript. Given my score is already positive, I shall keep it.

---

### Official Review · Reviewer_1FyB · 2024-11-02

**Soundness:** 3
**Presentation:** 4
**Contribution:** 4
**Rating:** 8
**Confidence:** 3

**Summary:**

This work challenges traditional binary views of machine-generated text detection for classifying entire text samples as either human or LLM-written, and instead studies detecting texts partially written by LLMs. The experiments consist of two main parts: a) estimating the percentage of LLM-written texts; b) identifying LLM-written segments.

**Strengths:**

1. The problem raised by this paper is very interesting and worth studying. This makes this paper have great academic impacts.
2. This paper overall is well-written. The presentation and structure are clear and concise, with well-crafted figures and only a few typos.
3. In-depth theoretical formulation and justification of the statistical method chosen.

**Weaknesses:**

1. Computational complexity: As the authors have acknowledged, scaling this method over many segments could be challenging. This poses limits to its applicability. In my opinion, this is a relatively minor issue as this partial detection problem is inherently harder, and we should encourage solutions to hard problems. This should be left for future works to work on.

2. Need more robustness analysis: My main criticism is that this method lacks more sensitivity analysis regarding to segment lengths and various forms of perturbation attacks. How would it perform when there are adversarial attacks trying to undermine the detection is another main problem we should consider. I understand the paper has limited scope and cannot study everything, but I would assign a more affirmative rating had the paper dedicated a section for robustness analysis.

3. This method also requires training with regards to the specialized domain, which makes it less generalizable. Although the authors have studied cross-domain performance, the results are less convincing as only two domains have been studied, and maybe domains with more distinct writing styles have more drastic performance difference. I point it out not because I want to ask the authors to do additional experiments on new domains (although you are welcomed to!), but it’s just that I think it is that the nature of the methods makes training necessary, and this weakness cannot be easily avoided.

4. A minor point: I would suggest to have the related work section review every existing work studying partially LLM-written text detection (I’ve searched for some and did not find too many), and reduce the part on watermarking methods as they are less relevant to this paper. A literature review in this understudied area would be very helpful!

**Questions:**

No further questions. Overall, I like the paper and would like to give positive ratings. Although there are imperfections, I think this work is valuable, memorable, and should be encouraged.

---

> ### Author Response · Authors · 2024-11-21
>
> We appreciate the reviewer's comments and recognition of our work. We hope the following responses help alleviate some of the reviewer's concerns.
>
> **Response to weaknesses**:
>
> **W1**. We agree that scaling the method is a relatively minor issue compared to the text identification task, and left for future work. Please refer to the section on complexity of PaLD-TI and greedy algorithm in the Initial Common Response regarding this topic.
>
> **W2**. Thank you for the suggestion. We are currently evaluating PaLD when it is faced with distribution shifts such as perturbations at inference time, and will update the manuscript shortly.
>
> **W3**. Regarding additional experiments, as mentioned in the Initial Common Response, we are currently evaluating several new domains from the RoFT and RAID datasets, and will add these results to the manuscript shortly.
>
> **W4**. Thank you for your suggestion. We will update the related works section with a few papers related to partial LLM-written text, and reduce the section on watermarking. To summarize, most such works detect the boundary when a text goes from being human-written to LLM-written. This means that each text only has two segments, where the first is human-written, and the second is LLM-written. The RoFT dataset [1], containing human-written sentences completed by GPT2, was created to evaluate how humans detect the boundary.
> [2, 3, 4] provide RoBERTa
> or Transformer-based models for solving the task in an automated fashion. [5] evaluates several
> approaches and finds that perplexity-based approaches are more robust for boundary detection. Our proposed framework, PaLD, is more general, as it encompasses settings where any segment of a multi-segment text could be human or LLM-written, does not require white-box access to an LLM, and can be used with both supervised $T$-scores as well zero-shot $T$-scores.
>
> **References**:
>
> [1] Liam Dugan, Daphne Ippolito, Arun Kirubarajan, and Chris Callison-Burch. RoFT: A tool for
> evaluating human detection of machine-generated text. In Qun Liu and David Schlangen (eds.),
> Proceedings of the 2020 Conference on Empirical Methods in Natural Language Processing:
> System Demonstrations, pp. 189–196, Online, October 2020. Association for Computational
> Linguistics. doi: 10.18653/v1/2020.emnlp-demos.25.
>
> [2] Joseph Cutler, Liam Dugan, Shreya Havaldar, and Adam Stein. Automatic detection of hybrid
> human-machine text boundaries, 2021.
>
> [3] Elizabeth Clark, Tal August, Sofia Serrano, Nikita Haduong, Suchin Gururangan, and Noah A. Smith.
> All that’s ‘human’ is not gold: Evaluating human evaluation of generated text. In Chengqing
> Zong, Fei Xia, Wenjie Li, and Roberto Navigli (eds.), Proceedings of the 59th Annual Meeting
> of the Association for Computational Linguistics and the 11th International Joint Conference
> on Natural Language Processing (Volume 1: Long Papers), pp. 7282–7296, Online, August
> 2021. Association for Computational Linguistics. doi: 10.18653/v1/2021.acl-long.565
>
> [4] Zijie Zeng, Lele Sha, Yuheng Li, Kaixun Yang, Dragan Gaševi´c, and Guanliang Chen. Towards auto-
> matic boundary detection for human-ai hybrid essay in education. arXiv preprint arXiv:2307.12267,
> 2023.
>
> [5] Laida Kushnareva, Tatiana Gaintseva, Dmitry Abulkhanov, Kristian Kuznetsov, German Magai,
> Eduard Tulchinskii, Serguei Barannikov, Sergey Nikolenko, and Irina Piontkovskaya. Boundary
> detection in mixed AI-human texts. In First Conference on Language Modeling, 2024

---

> > ### Comment · Reviewer_1FyB · 2024-11-21
> >
> > Thank you for the responses. I want to vouch for your paper, but since I cannot see some of the changes until you changed the manuscript, it becomes hard for me to raise much points. I will raise your soundness score though because your responses have affirmed my confidence in your paper.

---

> > > ### Author Response · Authors · 2024-11-21
> > >
> > > Thank you for raising the soundness score and we appreciate your desire to vouch for our work! We will inform you when the new manuscript is uploaded, and would be grateful if you could consider the score again by then.

---

> > > > ### Author Response · Authors · 2024-11-25
> > > >
> > > > We again thank the reviewer for their feedback. As described in the Second Common Response, we have now updated the manuscript to reflect the results shown in our initial point-by-point reply. For those comments, please refer to the main text (W1, W4) and the "Other Experimental Additions" section of the Second Common Response. In addition, we have provided new results to address comments from W2 and W3. These are summarized below:
> > > >
> > > > **W2**. Please see Sec. 4.3 (Table 7) regarding evaluation of PaLD when subject to paraphrasing attacks. We use the paraphrasing attacks used in Sadivasan et al (2023), that is, we use the T5-based Parrot model (Damodaran, 2021) to randomly paraphrase the LLM-written segments of the mixed texts. We randomly paraphrase 25\%, 50\% and 100\% of the LLM-written segments of the WP dataset, and evaluate the PE and TI tasks in Tab. 7. Shown below, all methods suffer a performance decrease with increasing attack strength. However, at all attack strengths, PaLD outperforms all other methods. We leave further analysis of mixed-text detector attacks for future work.
> > > >
> > > > Percentage estimation (MAE) and text identification (segment-wise accuracy) under paraphrase attacks on LLM-written segments in WP.
> > > >
> > > > | Fraction Chosen | PaLD-PE | RoBERTa-Reg | RoBERTa-QuantileReg | PaLD-TI | FastDetect-GPT-Seg | RoBERTa-Seg |
> > > > |----------------|----------|--------------|---------------------|----------|-------------------|-------------|
> > > > | 0.25 | 0.117 | 0.204 | 0.193 | 0.811 | 0.633 | 0.701 |
> > > > | 0.50 | 0.122 | 0.204 | 0.193 | 0.786 | 0.624 | 0.691 |
> > > > | 1.00 | 0.134 | 0.207 | 0.195 | 0.730 | 0.584 | 0.679 |
> > > >
> > > > **W3**. Regarding new datasets and new domains, we have evaluated RoFT and RAID in Sec. 4.2. This also contains new results on cross-LLM evaluation, where we evaluate our method on mixed-texts generated by Claude. Please refer to the Second Common Response for a summary of results.
> > > >
> > > >
> > > > References:
> > > >
> > > > Vinu Sankar Sadasivan, Aounon Kumar, Sriram Balasubramanian, Wenxiao Wang, and Soheil Feizi. Can ai-generated text be reliably detected? arXiv preprint arXiv:2303.11156, 2023.
> > > >
> > > > Prithiviraj Damodaran. Parrot: Paraphrase generation for nlu. Parrot: Paraphrase generation for nlu,
> > > > 2021.

---

> > > > > ### Author Response · Authors · 2024-11-28
> > > > >
> > > > > Dear reviewer 1FyB,
> > > > >
> > > > > Thank you once again for your review of our work. As the discussion period draws to a close, we would like to inquire if you had the chance to look at the updated manuscripts and results, have any additional questions, or found our responses to be helpful.
> > > > >
> > > > > Please let us know! We appreciate your time and feedback.

---

> > > > > > ### Author Response · Authors · 2024-12-02
> > > > > >
> > > > > > Dear reviewer 1FyB,
> > > > > >
> > > > > > Thank you for your service in reviewing our paper. As the discussion period draws to a close today, we would like to ask again if you had the chance to consider our previous response containing the updated manuscript and results, and whether it has addressed your concerns.
> > > > > >
> > > > > > Thanks!

---

> > > > > > > ### Comment · Reviewer_1FyB · 2024-12-03
> > > > > > > **Raised scores**
> > > > > > >
> > > > > > > Dear authors,
> > > > > > >
> > > > > > > Thank you for the updates. I've decided to raise my scores to 8 as my concerns are addressed.
> > > > > > >
> > > > > > > Best luck

---

> > > > > > > > ### Author Response · Authors · 2024-12-04
> > > > > > > >
> > > > > > > > Thank you! We appreciate your engagement during the discussion period, and for the constructive feedback that has improved the paper.

---

### Official Review · Reviewer_3hmG · 2024-11-03

**Soundness:** 2
**Presentation:** 3
**Contribution:** 3
**Rating:** 5
**Confidence:** 5

**Summary:**

The paper introduces Partial-LLM Detector (PaLD), a method designed to estimate what percentage of a text is generated by an LLM and identify which specific segments are machine-generated. This approach uses black-box methods leveraging scores from text classifiers to achieve these tasks. It offers a more nuanced detection capability compared to previous methods that only classify texts as either human or machine-generated in entirety.

**Strengths:**

1. This method allows for a more detailed understanding of the text composition by identifying LLM-generated segments.
2. The proposed method can handle texts of mixed origin, reflecting more realistic scenarios than those handled by traditional binary classifiers.
3. Provides statistically robust methods to back its detections, including confidence intervals and likelihood measures.

**Weaknesses:**

1. Some of the concepts and mathematical notations need further explanation for readers' better understanding.
2. The experimental setup needs improvement. Currently, experiments are conducted on only two specific datasets, so additional validation is required to assess generalizability. Moreover, the proposed method appears to rely on prior knowledge about distributional information, which could further limit its generalization ability.

**Questions:**

1. Line 93, could you please define the term 'robust' in a human-written-/LLM-text detection problem (e.g., what types of changes, uncertainties, and unexpected perturbations in input data are considered)?
2. Lines 90-100, Bayesian framework, MAP estimation, mixture Gaussian models, and greedy policy algorithm have already been widely used in text detection or related NLP problems, please provide some references to them.
3. Line 112, '..., and x \in \chi ...' \chi needs to be defined before the first time use it (e.g., is it a set or list).
4. Line 154, how do you determine the number of decomposed segments in real applications? Is it equal to the number of sentences in a paragraph?
5. Lines 190-193, are \eta and \delta different from each segment Xi? Or they are calculated over the entire mixed-text paragraph?
6. Lines 205-206, is \epsilon a pre-defined hyperparameter? Since it determines the range of p, will the optimal \epsilon depend on the distributions P_llm and P_human, and how to choose the optimal \epsilon?
7. The example in Fig. 6 only consists of 3 sentences. If the text has more sentences, will the T-score of all the combinations be computed? Or just 2-sentence combinations will be considered?
8. Lines 324-326, a greedy algorithm is employed to approximately solve the problem and reduce complexity. Could you briefly discuss the approximation errors introduced by this algorithm and whether or not they impact detection accuracy?
9. In section 4.2, the cross-domain detection experiment is particularly intriguing, and I can somewhat understand that this test is generally more challenging compared to in-domain detection. Could you please briefly describe whether the WP and Yelp data are correlated (or consist of some similar information) before we can safely reach the conclusion?
10. The experiments are conducted on only two specific datasets, so additional validation is required to assess generalizability. I am also curious about the performance of detection in more challenging cases, such as humans mimicking the LLM style or LLM imitating highly stylized human writing. These can be mentioned in future works or current limitations.

---

> ### Author Response · Authors · 2024-11-21
>
> Thank you for taking the time to read the paper and provide valuable feedback! Please find our responses to your comments in the following point-by-point response.
>
> **Response to weaknesses**:
>
> **W1**. Regarding clarity of the concepts and mathematical notations, we have addressed these in the responses to your questions below. We will also update the manuscript to reflect these changes. If there is still confusion, we are happy to help clarify specific concepts that are unclear to the reviewer.
>
> **W2**. Regarding additional experiments, as mentioned in the Initial Common Response, we are currently evaluating several new domains from the RoFT and RAID datasets, and will add these results to the manuscript shortly. Regarding distributional assumptions, there is (i) the mixed-text distributional assumption described around line 191, and there is also (ii) the conditional distribution of the $T$-scores given LLM fraction $\Delta$ that PaLD-PE uses, described in Sec. 3.2. For (i), this assumption of the underlying mixed-text data distribution simply says that each segment is either human or LLM, which is the only assumption that PaLD-TI makes. Note that this assumption is a novel assumption, and a contribution of our work. The mixture ratio $\eta$ is not assumed to be known. PaLD-PE in fact does not make any segment-level assumptions on the text $X$. For PaLD-TI, assuming each segment is either human or LLM is a general assumption that encompasses many scenarios of interest, such as those described in the introduction (e.g., LLMs editing parts of human-writtten articles, or humans refining a draft written by an LLM). For (ii), PaLD-PE fits a model to the $P(T(X)|\Delta)$ conditional distribution and assumes a Beta prior on $\Delta$, which is used to make inferences on the LLM fraction $\Delta$. Our ablation studies in Sec. 4.3 demonstrate that these distributional assumptions do not make a significant difference in performance. While a distribution shift in $P(T(X)|\Delta)$ could presumably occur due to different texts encountered, our cross-domain experiments in Sec. 4.2 demonstrate how a $P(T(X)|\Delta)$ distribution fit in one text domain still yields accurate predictions of the LLM fraction on another text domain.
>
>
> **Response to questions**:
>
> **Q1**. We thank the reviewer for pointing out this confusion. Classic LLM text detection methods, such as (Fast)DetectGPT and Ghostbuster, are designed to perform a binary classification on a paragraph which is either all LLM-generated or entirely human-written. However, in the proposed mixed-text setting, the size of the segment that are LLM-generated or human-written is usually shorter than a paragraph, e.g., only a sentence in a 10-sentence paragraph. This often causes a degradation of the performance of those classic LLM detection methods, as explained in line 94-95 and evidenced in Table 2 and 3. We will improve the wording and add more explanations to clarity in the manuscript.
>
> **Q2**. Thanks for the suggestion, and we will add references to Bayesian frameworks [1] and nonparameteric models in NLP [2] in the introduction in the updated manuscript shortly. Please also feel free to suggest any reference that is appropriate.
>
> **Q3**. In line 112, $\mathcal{X}$ is meant to be the sample space of texts that can be randomly drawn, such as all finite-length strings pertaining to a dictionary. Specifically, $\mathcal{X} = \mathcal{D}^*$, where $\mathcal{D}$ is a dictionary of characters, such as all ASCII symbols. We will update the manuscript accordingly to clarify this.
>
> **Q4**. In this work, we use sentences as segments for the running examples as well as the experiments, so the total number of segments would be the number of sentences of the text. This is a natural choice to make, as sentences are a natural way to segment text in human languages, and are thus a likely method humans may edit LLM written texts. As mentioned in Sec. 5, we leave a more general segmentation for future work.
>
> **Q5**. Regarding $\eta$ and $\delta$, please see the referenced footnote for an example at the beginning of Sec. 3. $\eta$ represents the fraction of segments that are LLM-written, and $\delta$ is the induced fraction of characters that are LLM-written. Given an $\eta$, a text $X$ is generated from the mixed-text distribution, and then $\delta$ can be calculated from $X$. For the 4-segment text at the beginning of Sec. 3, $\eta$ = 0.5 since LLM generates two sentences. However, there are 109 out of 197 LLM-generated characters, and therefore $\delta = 109/197 \approx 0.553$.

---

> ### Author Response · Authors · 2024-11-21
>
> **Q6**.  In Eq. (1), $\epsilon$ is not a hyperparameter, but rather a parameter of the normalized quantile slope set so that $1-\epsilon$ of the probability under the distribution of $T(X)$ is considered. In this way, it is a cutoff for the quantiles of $T(X)$, otherwise an infinite range of quantiles would need to be considered (i.e., $Q(\delta, p)=-\infty$ when $p=0$ and $Q(\delta, p)=\infty$ when $p=1$). Thus, we only consider the quantiles in the range of $p \in (\epsilon/2, 1-\epsilon/2)$ when computing $\sigma$. If $\epsilon$ is set too small, then many (potentially infinite) quantiles in the tails of the distribution (which are less significant) will need to be considered, as mentioned above. If $\epsilon$ is set too large, then not enough of the $T(X)$ distribution will be considered, and the estimated quantile slope may not reflect the full distribution of $T(X)$. In practice, when $\sigma$ is estimated, we set $\epsilon = 0.05$, so that 95\% of the quantiles of $T(X)$ are considered, and 5\% of the less relevant tails are ignored.
>
> **Q7**. The $T$-scores of all the combinations of segments will be considered, as implied by Eq. (4). Note that in this example each segment is a sentence. All subsets $S \subseteq \lbrace 1,\dots,n\rbrace$ are considered to maximize the $T$-score difference. Since we disregard $S = \emptyset$ and $S^\complement = \emptyset$ as mentioned in Sec. 3.3, the example in Fig. 6 shows all the subsets of $\lbrace 1,2,3\rbrace$ except $\emptyset$ and $\lbrace 1,2,3\rbrace$, which happen to be 1- and 2-sentence combinations. If the text has $k$ sentences, then PaLD-TI will consider 1- to $k-1$-sentence combinations.
>
> **Q8**. There are no theoretical guarantees on the approximation error that the greedy algorithm yields for our optimization problem in Eq. (4), as the objective function is unlikely to be submodular in $S$. Despite this, we experimentally show in Table 3 that PaLD-TI with greedy algorithm outperforms all other baselines by at least 9\% in terms of segment accuracy, with PaLD-TI with exact solver outperforming the baselines by at least 17\%. Please refer to the Initial Common Response for a more in-depth discussion regarding the complexity-performance tradeoff of PaLD-TI and greedy algorithm.
>
> **Q9**. The WP and Yelp datasets are from different domains and have different collection processes. They contain rather different text samples. WP's texts contain human-written short stories, and Yelp contains human-written reviews of restaurants, businesses, etc. This was in fact mentioned at the beginning of Sec. 4. In Table 9 (appendix), we show text samples (human and mixed-text), where the first two rows are examples from Yelp, and bottom two rows are examples from WP.
>
> **Q10**. Regarding additional experiments, as mentioned in the Initial Common Response, we are currently evaluating several new domains from the RoFT and RAID datasets, and will add these results to the manuscript shortly. Regarding more challenging cases such as human mimicking LLM styles, we are not aware of any current datasets that could support this experiment currently. We appreciate the suggestion and have mentioned these in the final remarks.
>
> **References**:
>
> [1] Shay Cohen. Bayesian Analysis in Natural Language Processing. Springer International Publishing,
> 2019.
>
> [2] Narges Sharif-Razavian and Andreas Zollmann. An overview of nonparametric bayesian models
> and applications to natural language processing, 2008.

---

> > ### Author Response · Authors · 2024-11-25
> >
> > We again thank the reviewer for their feedback. As described in the Second Common Response, we have now updated the manuscript to reflect the results shown in our initial point-by-point reply. For those comments, please refer to the main text (W1, Q1, Q2, Q3, Q8) and the "Other Experimental Additions" section of the Second Common Response. In addition, we have provided new results to address comments from W2, Q4, and Q10. These are summarized below:
> >
> > **W2** and **Q10**. Regarding new datasets and new domains, we have evaluated RoFT and RAID in Sec. 4.2 (Table 5 and 6). This also contains new results on cross-LLM evaluation, where we evaluate our method on mixed-texts generated by Claude. Please refer to the Second Common Response for a summary of results.
> >
> > **Q4**. As mentioned in our prior response to you, this paper assumes texts are segmented into sentences, which is a natural choice to make. We have added an experiment in Appendix A.3. to investigate the effect of a mismatched segmentation (Table 18). In other words, the case when the ground-truth segmentation (i.e., the actual segments where the text is fully LLM or fully human) does not perfectly align with the chosen segmentation of PaLD-TI. To do so, we use the same sentence-level mixed text, so the ground-truth segmentation is at the sentence-level. However, the chosen segmentation is now at every 2 sentences. This means that PaLD-TI (and segment baselines) can only assign prediction to pairs of sentences, and thus a misclassification (at the sentence-level) will occur whenever the two sentence within a pair differ in class (i.e., one is human and the other is LLM), no matter what the prediction is. We show how PaLD-TI and the FastDetectGPT-Seg perform below.
> >
> > | Chosen Segmentation | PaLD-TI | FastDetectGPT-Seg |
> > |---------------------|----------|-------------------|
> > | Every Sentence | 0.826 | 0.651 |
> > | Every 2 Sentences | 0.706 | 0.613 |
> >
> > As shown, there is a performance decrease in both PaLD-TI and FastDetectGPT-Seg, as expected, since perfect classification is no longer possible and mistakes are inevitable. We note that PaLD-TI still outperforms FastDetectGPT-Seg at the 2-sentence segmentation, and leave methods to mitigate mismatched segmentation for future work.

---

> > > ### Author Response · Authors · 2024-11-28
> > >
> > > Thank you once again for your review of our work. As the discussion period draws to a close, we would like to inquire if you have any additional questions or found our responses to be helpful.
> > >
> > > Please let us know! We appreciate your time and feedback.

---

> > > > ### Author Response · Authors · 2024-12-02
> > > >
> > > > Dear reviewer 3hmG,
> > > >
> > > > Thank you for your service in reviewing our paper. As the discussion period draws to a close today, we would like to ask again if you had the chance to consider our response, and whether it has addressed your concerns.
> > > >
> > > > Thanks!

---

### Official Review · Reviewer_DUtY · 2024-11-04

**Soundness:** 3
**Presentation:** 3
**Contribution:** 3
**Rating:** 6
**Confidence:** 4

**Summary:**

The authors introduce PALD-PE and PALD-TI, methods that estimate the percentage of LLM-generated content in a piece of text, and an approach to identifying the LLM-generated segments, respectively. PALD-PE provides MAP estimates of the percentage of LLM-generated text, and using gaussian kernel density estimation, they provide intervals that cover the ground-truth percentage. PALD-TI runs a combinatorial search over all possible segments using a score from a text-classifier to identify the LLM-generated segments. Furthermore, a greedy version of the PALD-TI approach is introduced. The authors evaluate their approach on the Writing Prompts and Yelp Dataset.

**Strengths:**

•	It is one of the first papers to address the more realistic mixed-text scenario and proposes to estimate the amount of text that is machine-generated, as well as to identify the segments that are machine-generated. This is a departure from the usual machine-or-human detection.

•	The paper includes rigorous ablations of the PALD-PE method, which greatly led credence to the observations that methods with a higher quantile slope were more useful in mix-text scenarios.

**Weaknesses:**

o	The fine-tuned RoBERTa baselines were trained to identify between fully human and fully-LLM texts that are larger than a single sentence in length, but then tested on individual sentences. This introduces a distribution mismatch during testing that might make the baselines weaker than they really should be.

o	For DetectGPT, and FastDetectGPT, the threshold should’ve been derived from a validation set to maximize their performance at the segment level identification task.

o	In Table 3, PALD TI relies on a supervised model, whereas DetectGPT and FastDetectGPT do not so it’s unfair to compare one against the other. It would be good to have a zero-shot version of PALD. It would’ve been interesting to test them against a version of PALD-TI which relies upon an unsupervised T-score.

o	The datasets and testing conditions considered are not sufficient. In particular, the work could’ve also been easily evaluated on the RoFT dataset (https://arxiv.org/abs/2212.12672), and it could’ve also been built off of RAID (https://arxiv.org/abs/2405.07940) and the M4 dataset (https://github.com/mbzuai-nlp/M4), both of which include more realistic testing domains.

o	Only GPT-4o was used when infilling. More LLMs should’ve been considered, including non-instruction-tuned LLMs.

•	The PALD-TI approach relies heavily on a T-Score from literature such as a trained RoBERTa model, FastDetectGPT, DetectGPT, etc. It uses them to perform a combinatorial search of 2^n – 2. A more efficient method for search that is not just greedy would make the method much stronger.

**Questions:**

•	Line 191, wrong notation? X_i instead of X?

•	Typo 297: performnace -> performance

•	What was the reason for having different fractions set for the testing split than for the training split?

•	Any insights as to why the quantile slope of DetectGPT is smaller, but the segment accuracy is stronger than that of GhostBuster and FastDetectGPT?

•	Were PALD-TI and PALD-PE evaluated on cases where there were no machine segments? How about in cases where there are only machine segments?

---

> ### Author Response · Authors · 2024-11-21
>
> Thank you for taking the time to read the paper and provide valuable feedback! Please find our responses to your comments in the following point-by-point response.
>
> **Response to weaknesses**:
>
> **W1**. Thank you for bringing the potential distribution mismatch to our attention. In fact, we choose to include the experimental results with RoBERTa trained on paragraph-level texts since the RoBERTa base model is pretrained on longer texts. Using sentence-level text is too short for the RoBERTa architecture.
> For the sake of completeness, we provide the RoBERTa classifier baselines trained on segment-length texts rather than paragraph-level texts, and report the resulting performance for both percentage estimation (PE) and and text identification (TI) tasks in the tables below. When compared to the baselines trained on paragraph-level data, their performance is actually worse (note that both training on paragraph-level and sentence-level do not perform well compared to other methods), despite the potentially better distributional alignment. We plan to update the manuscript with these results.
>
> Percentage estimation (MAE);
> RoBERTa classifiers trained on segment-level data.
>
> | Dataset | RoBERTa-Seg | RoBERTa-LN-Seg | PaLD-PE (ours; paragraph-level) |
> | -------- | ------- | ------- |----|
> | WP | 0.566 | 0.434 | 0.116 |
> | Yelp | 0.563 | 0.563 | 0.137 |
>
> LLM text identification
> (segment-wise accuracy); RoBERTa
> classifiers trained on segment-level data.
>
> | Dataset | RoBERTa-Seg | RoBERTa-LN-Seg | PaLD-TI (ours; paragraph-level) |
> | -------- | ------- | ------- | ---- |
> | WP | 0.498 | 0.502 | 0.826 |
> | Yelp | 0.496 | 0.496 | 0.801 |
>
> **W2**. Thank you for the suggestion. We have re-ran the DetectGPT and FastDetectGPT segment-wise baselines by setting the threshold to maximize the difference between TPR and FPR on a segment-level validation set.
> We show the difference by setting threshold at the paragraph level vs. sentence level in the tables below (it will be updated in Tab.~2, 3 in the upcoming revised version). There is an improvement in their performance, albeit around $<0.03$ in MAE for the percentage estimation task, and $<1\%$ for the segment-accuracy in the text identification task.
>
> Percentage estimation (MAE):
>
> | Dataset  |  DetectGPT (para.) | DetectGPT (sent.) |  FastDetectGPT (para.) | FastDetectGPT (sent.) | PaLD-PE (ours) |
> | -------- | ------- | ------- | ------- | ------- | ---|
> | WP       | 0.381    | 0.370 |0.212    | 0.184 | 0.116 |
> | Yelp     | 0.407    | 0.397 |0.218    | 0.190 | 0.137 |
>
> Text identification (segment accuracy):
>
> | Dataset  |  DetectGPT (para.) | DetectGPT (sent.) | FastDetectGPT (para.) | FastDetectGPT (sent.) | PaLD-TI (ours) |
> | -------- | ------- | ------- | ------- | ------- | ----- |
> | WP       | 0.486    | 0.488 | 0.650    | 0.651 | 0.826 |
> | Yelp     | 0.452    | 0.460 | 0.622    | 0.624 | 0.801 |
>
> **W3**. Thank you for your suggestions. Regarding unfairness of comparing zero-shot with supervised methods, in our paper, we are not restricting comparisons to only zero-shot methods, thus it is fair to compare against baselines that contain both zero-shot and supervised methods in Table 3. If one wishes to only compare among zero-shot methods, we indeed already have a zero-shot version of PaLD-TI, with PaLD-TI results using unsupervised T-score in the ablation study in Table 8, which reports PaLD-TI performance using a variety of T-scores. Comparing Table 3 and Table 8, we see that for each T-score used, PaLD-TI improves the text identification performance compared to the T-score segment-wise baseline; this improvement is most drastic for the RoBERTa, RoBERTa-LN, and DetectGPT T-scores, the last of which is zero-shot.
>
> **W4**. Regarding additional experiments, as mentioned in the Initial Common Response, we are currently evaluating several new domains from the RoFT and RAID datasets that you suggested, and will add these results to the manuscript shortly.

---

> ### Author Response · Authors · 2024-11-21
>
> **W5**. For the sentence-level mask-and-fill data generation, non-instruction-tuned LLMs do not perform well. For example, we found that Llama-3-8b without instruction tuning is not able to generate meaningful fills. As an example, when Llama-3-8b is prompted with our mask-and-fill prompt (Appendix A.1),
>
> ```
> What sentences should go in the 3 [MASK] locations of the following text? Only provide exactly one sentence per [MASK] location. Only provide the sentences as a numbered list with 3 sentences total.
> [MASK]. I got a taylor ham, egg, and cheese and a short stack of blueberry pancakes (I simply couldn’t decide between sweet or savory, plus leftovers are never a bad thing). The egg sandwich was fantastic. I was nervous when they said they didn’t have kaiser rolls so I went with a hamburger roll. It was an excellent decision. [MASK]. They layered the cheese on the sandwich, which to me is a must for a true egg sandwich. [MASK].
> ```
>
> The response is:
>
> ```
> 1. The blueberry pancakes were good, but not as good as the egg sandwich.
> 2. The egg sandwich was fantastic.
> 3. They layered the cheese on the sandwich, which to me is a must for a true egg sandwich.
> ```
>
> Similarly, we show an example using the Llama-3-8b-instruct:
>
> ```
> 1. The service was friendly and attentive, which is always a plus.
> 2. The egg sandwich was a bit messy, but that's just part of the charm.
> 3. The blueberry pancakes were fluffy and flavorful, with a perfect balance of sweet and tangy.
> ```
>
> Note that for the non-instruction-tuned Llama-3 model, fills 2 and 3 are copied and repeated from the original human-written text. Thus, to consider more LLMs for infilling, we are currently applying other instruction-tuned LLMs for evaluation of PaLD (see Table 21 for an example with the same prompt with GPT-4o), and will update the response soon. In addition, the new RoFT dataset contains human-LLM mixed text, where the LLM is GPT2-XL, which will provide another example for mixed text generated by another LLM other than GPT-4o.
>
> **W6**. Regarding the combinatorial search for PaLD-TI and other efficient methods, please refer to the Initial Common Response.
>
> **Response to questions**:
>
> **Q1**. The notation in the manuscript is correct. $X$ is the entire random mixed text, comprised of many segments. The $i$-th segment, $X_i$, is generated from the mixture distribution.
>
> **Q2**. Thank you for pointing out the typo, we will fix it in the updated manuscript.
>
> **Q3**. Actually, the fractions of the texts in the training split and testing split do match. Note that the target fractions for the testing split are 0.25, 0.5, and 0.75, but as described in Appendix A.1, this is not actually the true LLM text percentages, but rather the target fractions, which determine the number of masked sentences for the mask-and-fill procedure. Approximately 0.25, 0.5, and 0.75 of the sentences are masked out and filled by GPT-4o, then the true LLM fraction $\delta$ values are computed post-hoc. The true fractions depend on the text content and thus vary. We found that this procedure yielded similar distributions of $\delta$ over $[0,1]$ compared to setting the target fractions to 0.1, 0.2, ..., 0.9 as done for the training split. To provide concrete evidence for this, we compute a histogram of the $\delta$ values on the Yelp dataset for both train and test splits:
>
> Frequency of $\delta$ occuring in interval bins, on Yelp dataset:
>
> | Split | $[0,0.1]$ | $[0.1,0.2]$ | $[0.2,0.3]$ | $[0.3,0.4]$ | $[0.4,0.5]$ | $[0.5,0.6]$ | $[0.6,0.7]$ | $[0.7,0.8]$ | $[0.8,0.9]$ | $[0.9,1.0]$ |
> |-------| -------| -------| -------| -------| -------| -------| -------| -------| -------| -------|
> Train | 0.062 | 0.155 | 0.140 | 0.132 | 0.0972 | 0.0988 | 0.0957 | 0.106 | 0.0819 | 0.0315 |
> Test | 0.030 | 0.1367 | 0.170 | 0.1630 | 0.130 | 0.090 | 0.107 | 0.110 | 0.05330 | 0.01 |
>
>
> Thus, despite the ``target'' fractions differing between the train and test split, the true fractions actually match. In the upcoming revision, we will make this point clearer. Thank you for bringing it to our attention.
>
> **Q4**. We would like to emphasize that there can be a lot of noise in the estimation of the quantile slope, as it may depend on several sources of randomness such as the text distribution as well as the stochasticity of the $T$-scores (which is true for DetectGPT, Ghostbuster, and FastDetectGPT). Therefore, the quantile slope is only a proxy for the distinguishability of different $\delta$ values. As the quantile slopes for DetectGPT, Ghostbuster, and FastDetectGPT are the three smallest, and all around a similar value, their correlation with the resulting segment accuracy may be more influenced by randomness compared to the correlation between the higher quantile slope $T$-scores (e.g., RoBERTa) and their resulting segment accuracies. Further analysis on a good proxy, and also what properties a good detector should have, should be considered as an interesting future direction.

---

> > ### Author Response · Authors · 2024-11-21
> >
> > **Q5**. As the setting of this paper aims for mixed-text, we did not include the No LLM and the All LLM cases, as these two cases reduce to the classic setting of existing LLM text detection frameworks such as (Fast)DetectGPT and Ghostbuster.
> > On the WP dataset, we evaluate the PaLD framework on the No LLM and the All LLM (GPT-4o) cases and compare it with the FastDetectGPT, which outperforms DetectGPT and Ghostbuster in Table 2.
> > Note that FastDetectGPT is initially designed for the binary classification between No LLM and the All LLM.
> >
> > | Dataset | PaLD-PE (MAE) | FastDetectGPT-Seg (MAE) | PaLD-TI (Seg Acc.)| FastDetectGPT-Seg (Seg Acc.) |
> > |-----|-----|-----| ----- | ----- |
> > | No LLM | 0.137 | 0.261 | 0.900 | 0.829 |
> > | All LLM | 0.220 | 0.366 | 0.728 | 0.718 |
> >
> > The results further suggest that even on the classic setting, PaLD still provides competitive estimations. Note that PaLD-TI cannot pick all segments to be LLM or not LLM, and therefore only considers $2^n - 2$ of the $2^n$ possibilities. Thus, it will always make at least one mistake. This means the 0.900 segment accuracy for No LLM segments is the best it can achieve, since there are 10 segments per text in this dataset, and PaLD-TI will misclassify at least one segment incorrectly. Additionally, we would like to reiterate our response to Q3 that the testing datasets contain texts that are nearly all LLM or no LLM segments.

---

> > > ### Comment · Reviewer_DUtY · 2024-11-23
> > >
> > > Thank you very much for your thorough responses to each of my points, it is greatly appreciated. I'm willing to raise my score by one point, but would like to wait until the results on the RoFT and RAID datasets are complete.

---

> > > > ### Author Response · Authors · 2024-11-25
> > > >
> > > > We again thank the reviewer for their feedback. As described in the Second Common Response, we have now updated the manuscript to reflect the results shown in our initial point-by-point reply. For those comments, please refer to the "Other Experimental Additions" section of the Second Common Response. In addition, we have provided new results to address comments from W4 and W5. These are summarized below:
> > > >
> > > > **W4**. Regarding new datasets and new domains, we have evaluated RoFT and RAID in Sec. 4.2 (Table 5 and 6). Please refer to the Second Common Response for a summary of results.
> > > >
> > > > **W5**. Regarding additional LLMs, we have evaluated Claude-3.5-Sonnet in Sec 4.2 (Table 5 and 6). Please refer to the Second Common Response for a summary of results.

---

> > > > > ### Comment · Reviewer_DUtY · 2024-11-25
> > > > >
> > > > > Thank you for your responses, I've raised my score to a 6.

---

> > > > > > ### Author Response · Authors · 2024-11-26
> > > > > >
> > > > > > Thank you! We appreciate your engagement during the discussion period, and for the constructive feedback that has improved the paper.

---

### Author Response · Authors · 2024-11-21
**Initial Common Response to Reviewers**

We appreciate the reviewers taking the time to read our paper and provide helpful comments and feedback. We thank the reviewers for recognizing the impactfulness of PaLD, as it is the first work to initiate study of and provide solutions for the more realistic mixed-text LLM detection setting. In this common response, we first address the concern about more experiments and domains made by multiple reviewers. To address the reviewers' individual concerns, we have responded to each reviewer's comments below with a point-by-point reply, and hope they help alleviate the reviewer's concerns. The manuscript will be updated soon to reflect suggested changes.

**New experiments and domains**: A common feedback from all reviewers was to provide more experimental results on datasets or domains aside from the WritingPrompt (WP) and Yelp datasets. As suggested by reviewer DUtY, we are currently evaluating our method on the RoFT and RAID datasets, and plan to update this thread and the manuscript with results once they are complete. The RoFT dataset contains texts from domains such as recipes, New York Times articles, and short stories. The mixed-texts are generated by prompting GPT2-XL to complete $10-k$ sentences given the first $k$ human-written sentences as an input prompt. The RAID dataset contains domains such as paper abstracts and Wikipedia articles. RAID does contains mixed texts, so we apply the same mask-and-fill approach described in the manuscript (cf. Appendix A.1).

**Complexity of PaLD-TI and greedy algorithm**: Another concern brought up by several reviewers is about the computational efficiency of PaLD-TI. As mentioned in the paper, it is a combinatorial problem (actually NP-hard) with complexity exponential in the number of segments $n$. The greedy algorithm already significantly reduces runtime for texts with $n$ segments, from $O(2^n)$ to $O(n^2)$, as the greedy method requires $O(n)$ complexity per iteration. Additionally, greedy has no hyperpameter choices, is straightforward to implement, and unlike many other approximate methods, is amenable to parallel batch computation, as computing $f(S \cup e) - f(S)$ over all $e \in A$ in Alg. 1 can be batched together. As experimentally verified in Table 3, PaLD-TI with greedy already outperforms all other baselines by at least 9\% in terms of segment accuracy. Determining if other methods may achieve a better performance-complexity tradeoff compared to the greedy algorithm could be research left for future work, as the current paper focuses on formulations for solving the text identification problem itself. Regarding performance guarantees, another potential benefit of greedy is that it returns a $(1-1/e)$ approximation of the optimal when the objective is submodular. Additionally, solving PaLD-TI exactly is reasonably efficient for texts of up to 10 segments, which is what is considered in this paper. To give concrete running times on a single A10 GPU, on average, running PaLD-TI with exact solver takes $\approx 30$ seconds for a paragraph-level text of 10 segments (around 200 words), whereas running PaLD-TI with greedy solver takes $\approx 2.1$ seconds; this is with the RoBERTa-based $T$-scores. If a text with many segments is encountered in practice, a more practical method may be to divide the text into chunks, and run greedy/exact on those chunks. For a chunk size $b$ on $n$ segments, solving PaLD-TI exactly would have complexity $O((n/b) 2^b)$, and with greedy $O((n/b) b^2)$. We plan to update the manuscript with these discussions.

---

### Author Response · Authors · 2024-11-25
**Second Common Response to Reviewers**

Again, we thank all the reviewers for their time in providing us with helpful feedback and interactions. We have now **updated the manuscript**, with edited content highlighted in blue.

## Experiments on new domains and LLMs

Regarding the common feedback from all reviewers to provide more experimental results on additional domains, we have added the following:

- Additional results on mixed-texts written by a different LLM (Claude-3.5-Sonnet), in Sec. 4.2, Table 6 (reviewer DuTY)
- Additional experiments on the RAID and RoFT datasets, a total of 5 new domains, shown in Sec. 4.2, Tables 5 and 6. These datasets are used to both further support the cross-domain generalization performance, as well as cross-LLM generalization performance (all reviewers)


We summarize the results here.

**Fixed-LLM**: We first keep the LLM that generates the mixed-texts as GPT-4o, and evaluate PaLD when the data domain during test time shifts. For PaLD-PE, the likelihood fitting phase is performed on the WritingPrompts (WP) dataset, and the point estimate or predictive interval is estimated on the testing dataset of a different domain;
similarly, the baselines are trained on WP and evaluated on a different testing domain. The testing domain datasets we evaluate on include Yelp and RAID. For RAID, we use the Abstracts and Wiki domains. For text identification, PaLD-TI does not require training, but may use a $T$-score that is fitted to a dataset (e.g., RoBERTa-LN trained on WP). The FastDetectGPT baseline (applied segment-wise) has threshold set on WP, and RoBERTa (segment-wise) is trained on WP. The results are shown here:

Cross-domain fixed-LLM results for percentage estimation (mean absolute error) and text identification (segment-wise accuracy). Training dataset: WP; LLM: GPT-4o.

| Test Dataset | PaLD-PE | RoBERTa-Reg | RoBERTa-QuantileReg | PaLD-TI | FastDetect-GPT-Seg | RoBERTa-Seg |
|--------------|---------|-------------|--------------------|---------|--------------------|--------------|
| Yelp | **0.147** | 0.211 | 0.231 | **0.801** | 0.626 | 0.592 |
| Abstracts | **0.151** | 0.306 | 0.223 | **0.718** | 0.693 | 0.656 |
| Wiki | **0.224** | 0.313 | 0.238 | 0.623 | 0.636 | **0.735** |

PaLD-PE remains the best-performing method compared to the baselines. This demonstrates that PaLD-PE has a superior ability to generalize across domains,
We see that PaLD-TI is superior to baselines on Yelp and Abstracts but not Wiki, where RoBERTa-Seg achieves high segment-wise accuracy.

**Cross-LLM**: Then, we evaluate the case when the LLM that generated the mixed-text may also be different at test time. We use a similar setup to the previous section. In particular, we evaluated PaLD and baselines trained/fitted on WP (GPT-4o-written) with the testing domains set as WP, Yelp, and RAID (Abstracts, Wiki) generated using Claude-3.5-Sonnet. We also evaluate RoFT, which are mixed texts where the first $k$ sentences are human, and remaining $10-k$ are GPT2-written; we use the Short Stories (SS), Recipes, and New York Times (NYT) domains. Note all are cross-domain and cross-LLM except for WP, which is only cross-LLM.

| LLM | Test Dataset | PaLD-PE | RoBERTa-Reg | RoBERTa-QuantileReg | PaLD-TI | FastDetect-GPT-Seg | RoBERTa-Seg |
|-------|--------------|---------|-------------|--------------------|---------|--------------------|--------------|
| Claude | WP | **0.114** | 0.184 | 0.138 | **0.751** | 0.531 | 0.675 |
| Claude | Yelp | **0.128** | 0.193 | 0.144 | **0.731** | 0.628 | 0.654 |
| Claude | Abstracts | **0.115** | 0.204 | 0.145 | **0.710** | 0.633 | 0.651 |
| Claude | Wiki | **0.174** | 0.245 | 0.185 | 0.602 | 0.634 | **0.683** |
| GPT-2 | SS | **0.227** | 0.236 | **0.227** | 0.533 | **0.610** | 0.525 |
| GPT-2 | Recipes | **0.166** | 0.209 | 0.186 | **0.548** | 0.520 | 0.505 |
| GPT-2 | NYT | **0.236** | 0.246 | 0.239 | 0.477 | **0.613** | 0.544 |

On Claude, PaLD-PE outperforms all baselines despite the domain and LLM shift. PaLD-TI is superior on all domains except Wiki, where RoBERTa-Seg again achieves high accuracy. For GPT-2 (i.e., RoFT data), PaLD-PE performs similar or better. For the TI task, both PaLD-TI and RoBERTa-Seg result in a significant performance decrease compared to FastDetectGPT; this may be due to poor generalization to GPT-2 texts of the RoBERTa-based classifiers that is used for both RoBERTa-Seg and the $T$-score for PaLD-TI. The better generalization to Claude from GPT-4o compared to GPT-2 may be expected, as the parameter count of GPT-2 is orders of magnitude lower compared to Claude and GPT-4o.

---

> ### Author Response · Authors · 2024-11-25
>
> ## Other Experimental Additions
>
>
> In addition to the evaluations on new LLM data and new domains, we have updated the manuscript with the following:
>
> - Reviewer DuTY: A comparison on the distribution of the ground-truth LLM fraction $\delta$ between the training and testing split of Yelp, demonstrating that although the _target_ fractions differ, the actual character-level LLM fraction distribution is the same. This is shown in Appendix A.1 (Table 9).
> - Reviewer DuTY: An ablation study showing the ineffectivness of finetuning the RoBERTa segment-wise baselines on segment-level data, shown in Appendix A.3 (Tables 10, 11).
> - Reviewer DuTY: Updated results on DetectGPT and FastDetectGPT baselines, where the threshold chosen is now based on segment-level validation data. This is shown throughout the results section (Tables 2 and 3), and a direct comparison between thresholding based on segment-level vs. paragrpah-level data is shown in Appendix A.3 (Tables 19, 20).
> - Reviewer DuTY: Results on how PaLD performs on text consisting of all-LLM or no-LLM segments in Appendix A.3 (Tables 16, 17).
> - Reviewer 1FyB: An added section in the related works discussing partial LLM-written text, namely LLM boundary detection
> - Reviewer 1FyB: Evaluation of PaLD when subject to paraphrasing attacks, shown in Sec. 4.3 (Table 7).
> - Reviewer qnTJ: An ablation study on the KDE bandwidth for PaLD-PE, shown in Appendix A.3 (Table 13).
> - Reviewer qnTJ: Results on segment-level $F_1$ scores for LLM text identification in Appendix A.3, providing more context on misclassifications (Table 12).
> - Reviewer qnTJ: A result demonstrating the effect of segmentation mismatch in Appendix A.3. (Table 18).
> - Reviewers DUtY and 3hmG: Minor typos or clarification throughout the text
> - Reviewers DUtY, 3hmG and 1FyB: Discussion on practical runtimes of PaLD-TI with exact and greedy solver (Section 4.1).
>
> We hope that these new updates help alleviate the reviewers' concerns about our work.

---

### Meta-Review · Area_Chair_bCcc · 2024-12-08

**Metareview:**

The paper introduces Partial-LLM Detector (PaLD), a method designed to address the mixed-text scenario, where texts are partially generated by LLMs and partially written by humans. The proposed methods, PALD-PE and PALD-TI, estimate the percentage of LLM-generated content and identify specific LLM-generated segments, respectively. The authors evaluate their methods on the Writing Prompts and Yelp datasets and propose improvements such as a greedy combinatorial search to enhance scalability.

This paper makes a great contribution to the field by tackling the mixed-text problem, an important yet underexplored area in LLM detection. The methods are innovative, the experiments are comprehensive, and the results are promising. While there are areas for improvement, such as dataset diversity and computational scalability, the strengths of the paper outweigh its weaknesses. I recommend acceptance with minor revisions to address these concerns in future iterations.

**Additional Comments On Reviewer Discussion:**

Additional experiments were asked for by the reviewers including new domains and more LLMs, as well as parameter analyses, which have been properly addressed by the authors. All reviewers are happy about the new results, except for one not respond to the discussion.

---

### Decision · Program_Chairs · 2025-01-22

Accept (Poster)